# Large-scale genetic analysis reveals mammalian mtDNA heteroplasmy dynamics and variance increase through lifetimes and generations

Joerg P. Burgstaller[1,2,3], Thomas Kolbe[4,5], Vitezslav Havlicek[6], Stephanie Hembach[1], Joanna Poulton[7], Jaroslav Piálek[8], Ralf Steinborn[9], Thomas Rülicke[10], Gottfried Brem[1,2], Nick S. Jones[3,11] & Iain G. Johnston [12]

Vital mitochondrial DNA (mtDNA) populations exist in cells and may consist of heteroplasmic mixtures of mtDNA types. The evolution of these heteroplasmic populations through development, ageing, and generations is central to genetic diseases, but is poorly understood in mammals. Here we dissect these population dynamics using a dataset of unprecedented size and temporal span, comprising 1947 single-cell oocyte and 899 somatic measurements of heteroplasmy change throughout lifetimes and generations in two genetically distinct mouse models. We provide a novel and detailed quantitative characterisation of the linear increase in heteroplasmy variance throughout mammalian life courses in oocytes and pups. We find that differences in mean heteroplasmy are induced between generations, and the heteroplasmy of germline and somatic precursors diverge early in development, with a haplotype-specific direction of segregation. We develop stochastic theory predicting the implications of these dynamics for ageing and disease manifestation and discuss its application to human mtDNA dynamics.

[1] Department for Agrobiotechnology, Biotechnology in Animal Production, IFA Tulln, 3430 Tulln, Austria. [2] Institute of Animal Breeding and Genetics, University of Veterinary Medicine Vienna, Veterinärplatz 1, 1210 Vienna, Austria. [3] Department of Mathematics, Imperial College London, London SW7 2AZ, UK. [4] Biomodels Austria, University of Veterinary Medicine Vienna, Veterinaerplatz 1, 1210 Vienna, Austria. [5] University of Natural Resources and Life Sciences, Konrad Lorenz Strasse 20, 3430 Tulln, Austria. [6] Department for Biomedical Sciences, Reproduction Centre Wieselburg, University of Veterinary Medicine, Vienna, Austria. [7] Nuffield Department of Women's and Reproductive Health, University of Oxford, Oxford, United Kingdom. [8] Research Facility Studenec, Institute of Vertebrate Biology of the Czech Academy of Sciences, Květná 8, 603 65 Brno, Czech Republic. [9] Genomics Core Facility, VetCore, University of Veterinary Medicine Vienna, Veterinärplatz 1, 1210 Vienna, Austria. [10] Institute of Laboratory Animal Science, University of Veterinary Medicine Vienna, Veterinärplatz 1, 1210 Vienna, Austria. [11] EPSRC Centre for the Mathematics of Precision Healthcare, Imperial College London, London SW7 2AZ, UK. [12] School of Biosciences, University of Birmingham, Birmingham B15 2TT, UK. Correspondence and requests for materials should be addressed to J.P.B. (email: joerg.burgstaller@vetmeduni.ac.at) or to N.S.J. (email: nick.jones@imperial.ac.uk) or to I.G.J. (email: i.johnston.1@bham.ac.uk)

Mitochondrial DNA (mtDNA) exists in large copy numbers in most eukaryotic cells, and encodes functionally vital parts of bioenergetic machinery. Mutations and gene therapies lead to different mtDNA sequences present in the same cell: the population fraction of a non-wildtype mtDNA in a cell is termed heteroplasmy[1]. The cell-to-cell mean and variance of heteroplasmy dictate the inheritance and onset of deadly mitochondrial diseases, but how these quantities change with time and through generations is poorly understood[2]. Cutting-edge gene therapies aiming to prevent mitochondrial disease may be challenged if mean heteroplasmy changes over time[3–5], and changes in cell-to-cell heteroplasmy variability over time and between generations influence the probabilities with which mitochondrial diseases become manifest and the success of therapeutic strategies[4]. However, technological and ethical limitations mean that the dynamics of these populations are hard to observe, especially in humans, challenging both our understanding of fundamental biology and our ability to optimise therapies.

In particular, the cell-to-cell variance of heteroplasmy over organismal lifetimes remains poorly understood, despite its importance both for mtDNA diseases and for fertility strategies. Higher heteroplasmy variance increases the probability that a threshold heteroplasmy is crossed by cells, a requisite for disease manifestation[6]. On the other hand, higher variance also increases the probability of cells having low heteroplasmies. This is desirable in pre-implantation genetic diagnosis (PGD), a therapeutic approach aiming to address the inheritance of heteroplasmic mitochondrial disease[4]. In PGD, several embryos from a carrier mother are sampled for heteroplasmy before they are implanted. These set of embryos will typically have a range of heteroplasmy values – those with lowest measured heteroplasmy will be selected for implantation. Clearly in this situation, high heteroplasmy variance is desirable: the wider the spread of heteroplasmies, the greater the probability that at least one embryo will have a low

heteroplasmy and will be suitable for implantation[7]. However, our lack of knowledge about the features governing heteroplasmy variance represents a comparative blind spot in our ability to optimise clinical advice. In particular, the influence of maternal age – a central consideration in fertility treatments – remains unclear. Modelling and modern statistical approaches are beginning to shed light on processes underlying mtDNA dynamics through development and ageing[3,8]; however, the limited scale of existing datasets has limited our ability to elucidate the dynamics and timescales of these processes, particularly in the case of heteroplasmy variance, which requires large sample sizes to characterise[8,9].

The characterisation of heteroplasmy variance over time requires the disambiguation of the set of stochastic processes that may modulate it. A process known as the mtDNA bottleneck acts to increase heteroplasmy variance during development[1,7,10,11]. This increase in variance allows a circumvention of Muller's ratchet (the ongoing buildup of deleterious mutations) by segregating mutation load across cells, and hence allowing the selection of lower-heteroplasmy cells. The mechanism and timing of the mtDNA bottleneck has been debated, but stochastic modelling has shown that several of these competing hypotheses are compatible with the induction of variance through a combination of random partitioning and ongoing replication and degradation of mtDNA molecules[2,7,12]. This random turnover of mtDNA, leading to drift in heteroplasmy[7,13–15], also occurs throughout ageing, but its dynamics remain hard to characterise. These bottleneck and drift components are often coarse-grained together into a single effective bottleneck[13,16–19]. Flexibility in the developmental bottleneck[7] and differences in the amount of drift (potentially due, for example, to differences in age[20], or in physical dynamics of mitochondria[21]) can then both be responsible for heterogeneity in this effective bottleneck size, making it hard to dissect the influence of ageing and development on vital mtDNA statistics.

To address this challenging lack of quantitative understanding, we set out to explore how maternal age affects heteroplasmy

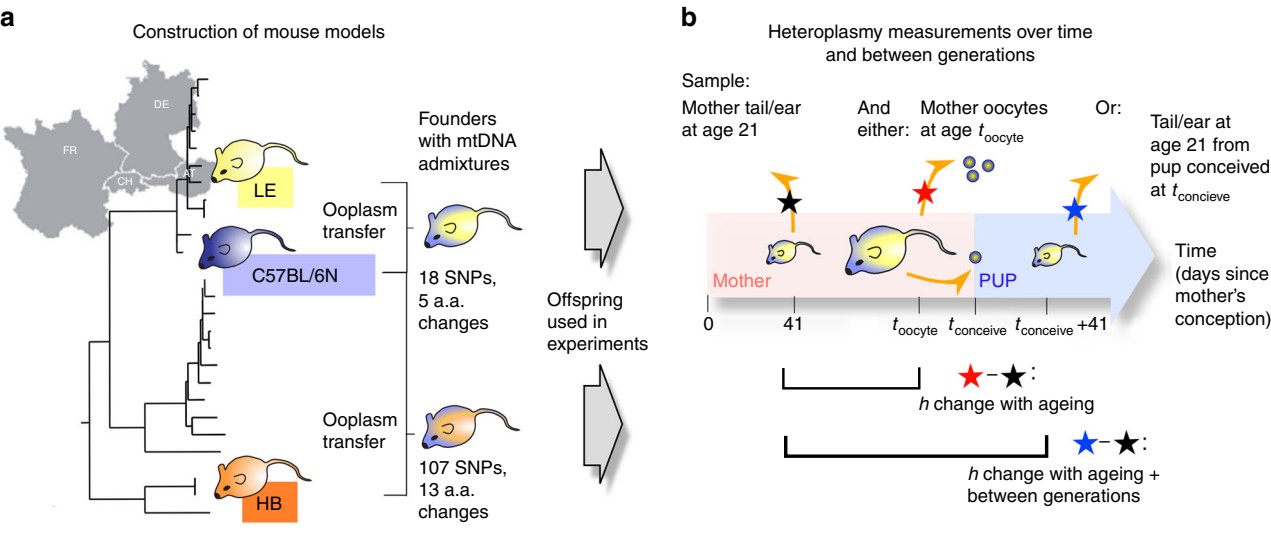

**Fig. 1** Analysing mtDNA heteroplasmy in genetically distinct wild-derived mouse models. **a** Two mouse strains, LE and HB, were selected from a diverse set of mice captured across central Europe and sequenced. LE and HB display different degrees of mtDNA relatedness to lab mouse strain C57BL/6N. Mouse models with admixed mtDNA populations were created using ooplasm transfer, yielding founder females with a wide range of heteroplasmies of LE and HB mtDNA, all on a C57BL/6N nuclear background. **b** Offspring of these founders are used for heteroplasmy sampling. A reference tissue (tail or ear) is sampled from mice at 41 days after conception (black star – 21 days post-partum). For some mice (red star), single oocyte heteroplasmies are then measured at a later time $t_{oocyte}$. For others (blue star), mice are allowed to conceive (which occurs at time $t_{conceive}$) and bear litters; the pups in these litters are sampled for heteroplasmy at age 21 days post-partum. We thus characterise heteroplasmy changes, relative to a reference tissue, in germline over organismal ageing (red star vs. black star), and over ageing and between generations (blue star vs. black star)

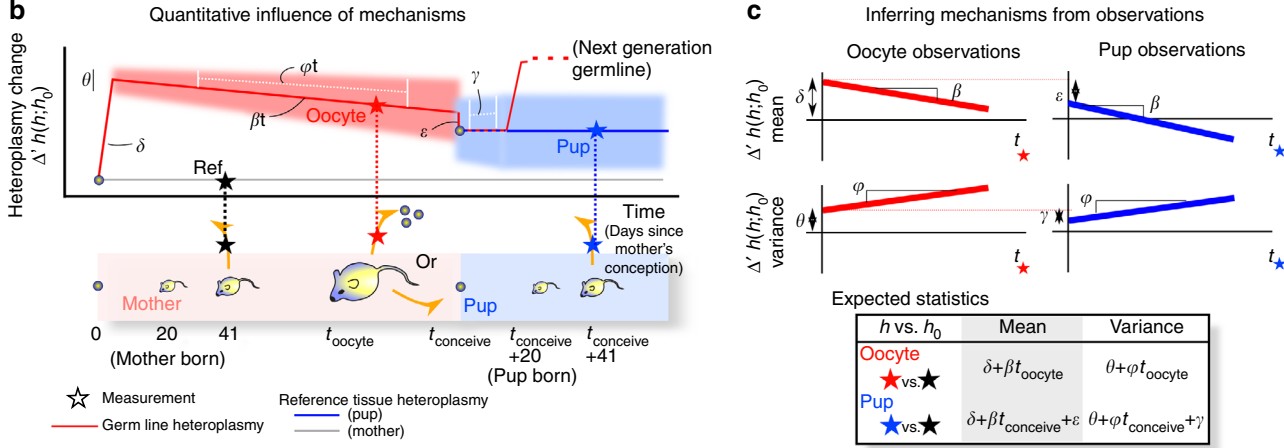

**Fig. 2** Inference of processes affecting heteroplasmy statistics through life and generations. **a** The set of processes that potentially influence heteroplasmy, inducing both one-off and time-dependent changes in mean and variance. **b** Illustration of the influence of these processes on observed mtDNA statistics (transformed change from reference tissue) over time. Solid lines give mean heteroplasmy; shaded regions denote variance, both of which may vary over time according to the corresponding parameter. **c** Relating the observations from our dataset construction protocol (Fig. 1) to the values of these mechanistic parameters, allowing us to use parameteric inference to identify and learn the contribution of each mechanism to observe heteroplasmy patterns. Supplementary Figure 1 illustrates the expected behaviour of mtDNA observations for further different values and combinations of these mechanistic parameters

statistics in mammals. Here, we produce and use an mtDNA dataset of unprecedented scale, linking mothers and offspring in mouse models containing admixtures of lab and wild-derived mtDNA, representing a controlled degree of population diversity[3] (Fig. 1). We design and use a bottom-up mathematical model in conjunction with statistical inference to characterise the dynamics of admixed mtDNA populations as organisms age and produce offspring. Our data allows us to quantitatively characterise six variables that together govern both heteroplasmy shift and variance between generations.

We find that mtDNA heteroplasmy variance in oocytes and offspring increases linearly with maternal age, and characterise this increase in unprecedented quantitative detail. We analyse heteroplasmy shifts between mother and her oocytes and pups both in a neutral and a non-neutral segregation model, and find that heteroplasmy variance is strongly influenced by both mtDNA composition and tissue type. Moreover, we pinpoint the heteroplasmy shift in the non-neutral model to early development, and formulate a method, with potential for clinical

application, for predicting the risk of embryonic heteroplasmy exceeding pathogenic thresholds, given maternal age. Our findings provide new insights into mtDNA inheritance and disease manifestation, which may be leveraged to optimise human reproductive techniques.

## Results
**Large-scale sampling of single oocyte and pup heteroplasmy.** Figure 1 illustrates the data sampling approach in our study. Two genetically distinct mouse models, LE and HB, are used. Both contain an admixture of a wild-derived mtDNA type from central Europe and C57BL/6N mtDNA. The labels refer to the German localities Lehsten and Hohenberg, where the original wild mice were captured[3]. The LE mtDNA is closely related to the C57BL/6N mtDNA (only 18 SNPs difference), while the HB mtDNA is genetically more distant with a difference of 107 SNPs[3]. Reference heteroplasmy samples were taken from tail or ear tissue, according to animal welfare requirements, at the consistent age of 21 days. Tail and skin (comparable to ear) tissue in our models

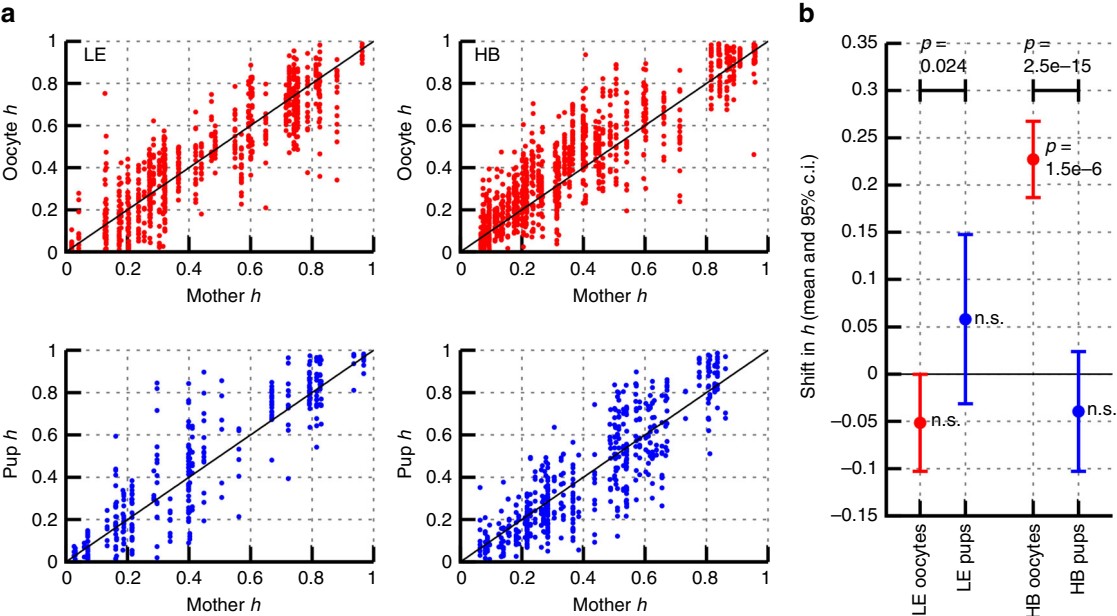

**Fig. 3** MtDNA heteroplasmy shift between mother and oocytes/offspring. **a** Heteroplasmy in mother reference tissue (x-axis) against the heteroplasmy values of their respective oocytes or pups (y-axis). Circles indicate individual measurements of single oocytes; or reference tissue biopsies of pups at day 21 post-partum. The diagonal line marks theoretical neutral segregation. **b** Summaries of heteroplasmy shifts relative to reference tissues. P-values associated with datapoints give the result of a Mann–Whitney test against the null hypothesis of both reference and subsequently measured heteroplasmies having the same mean. P-values associated with comparisons (top) give the result of a Mann–Whitney test against the null hypothesis that oocyte and pup heteroplasmies have the same mean. In the LE model, segregation is close to neutral in oocytes and pups: heteroplasmy values of both oocytes and pups average around that of their mothers, with the expected variation caused by the bottleneck. In the HB model, a clear increase of HB mtDNA is visible in the oocytes, which is reversed in offspring. The direction of shift from oocyte to pup is different in the two different mouse models: heteroplasmy increases in LE and decreases in HB. Oocytes: $n = 795$ LE; 1152 HB. Pups: $n = 552$ HB, 347 LE

display negligible relative and absolute heteroplasmy shifts, particularly at this early age[3,22]. For some mice, single oocyte heteroplasmy measurements were taken at a subsequent time point $t_{oocyte}$. For others, mice are allowed to conceive and produce pups, and tail or ear samples are taken from these pups at age 21 days. As a broad sampling of heteroplasmy behaviour across ages is desirable to facilitate characterisations of the mechanisms governing mtDNA populations[8], we took oocyte measurements from mice of ages 3–388 days, and from pups from mothers of age 60–329 days. We obtained 795 LE oocyte samples, 1152 HB oocyte samples, 347 LE pup samples, and 552 HB pup samples, representing a spread of heteroplasmies from 0 to 100%. In LE, a mean of 19 (s.d. 6.4) oocyte samples were taken from each of 43 females, and a mean of 6.6 (s.d. 2.8) pup samples were taken linked to 56 distinct mothers. In HB, a mean of 21 (s.d. 6.8) oocyte samples were taken from each of 56 females, and a mean of 7.3 (s.d. 2.7) pup samples were taken linked to 77 distinct mothers. All heteroplasmy samples are available as Supplementary Data 1.

**Umbrella stochastic model unifying mtDNA dynamics.** To facilitate an unbiased and statistically powerful analysis of our large-scale genetic data, we first construct a quantitative platform using a family of parameters to describe the influence of a set of possible processes that potentially affect mtDNA population statistics. We initially assume nothing about these influences: each parameter may be zero (indicating that the corresponding process has no influence on mtDNA) or take a nonzero value (indicating the presence of an influence and quantifying its effect). The strength of coupled modelling and inference approaches to harness and unify mtDNA

observations has been demonstrated across tissues in mice[3] and in human ontogenic phylogenies in ref. [8]. Our stochastic platform allows us to infer the possible values of each parameter, and thus the influence of each process, harnessing our full dataset for each genetic pairing.

The set of processes that we consider which could conceptually influence heteroplasmy statistics from mother to pup are listed, each with a corresponding parameter, in Fig. 2a. The potential influence of each on the mean or variance of cell-to-cell heteroplasmy levels is illustrated in Fig. 2b and some examples of the range of possible behaviours under this umbrella model are given in Supplementary Figure 1, with illustrations of the corresponding biological observations that would be expected in each case.

The combination of the influence of these processes, in conjunction with our heteroplasmy transformation (see Methods section, Eq 4), leads to the following expected distributions for the transformed heteroplasmy observations $\Delta h'$ in our dataset, for the oocyte data and the pup data respectively:

$$\Delta' h(h_{oocyte}(t); h_0) \sim N(\delta + \beta t, \ \theta + \varphi t) \qquad (1)$$

$$\Delta' h(h_{next\ gen\ ref}(t); h_0) \sim N(\delta + \beta t + \varepsilon, \ \theta + \varphi t + \gamma), \qquad (2)$$

where $N(\mu, \sigma^2)$ is the normal distribution with mean $\mu$ and variance $\sigma^2$. This model structure (Fig. 2b) underlies our analysis of these genetic data. We will proceed by analysing the evidence for individual processes within this model: we do not at first directly infer the parameters, but instead construct summary statistics which capture their core properties (Figs 3–5). Having explored our data and observed clear summaries of the

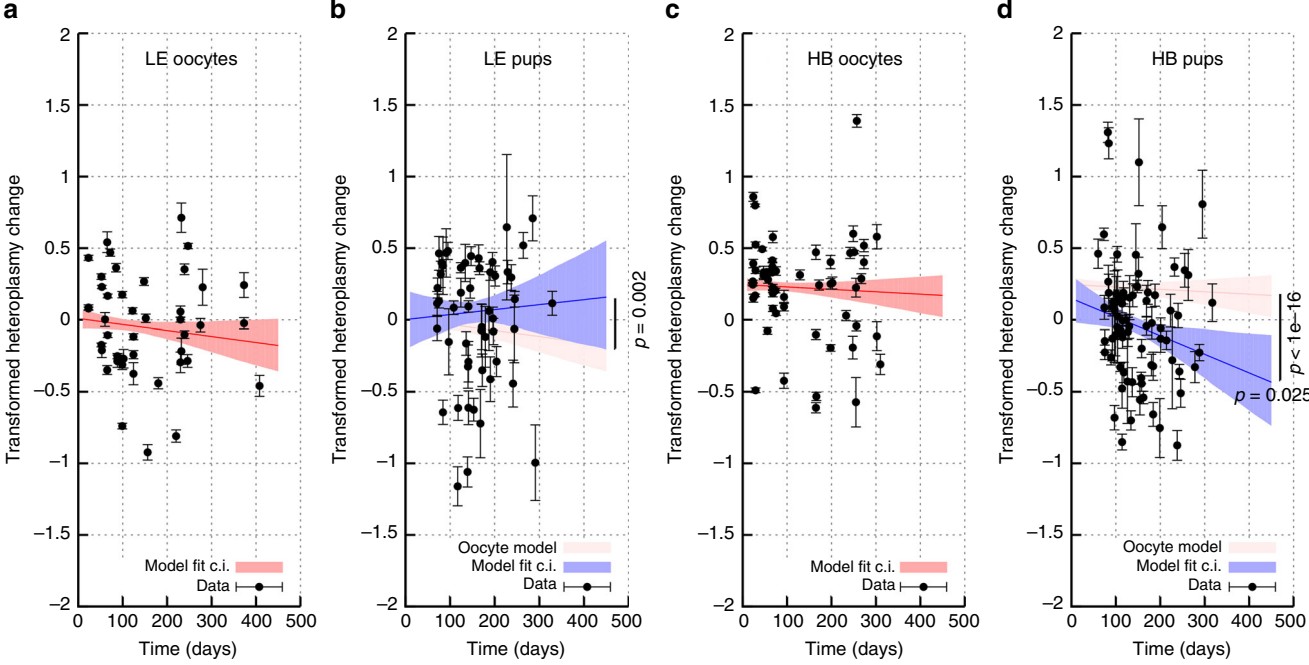

**Fig. 4** Limited change in mean heteroplasmy with maternal age in oocytes and pups. Transformed heteroplasmy change (Eq. 4) between reference measurement and oocyte or pup samples in LE (**a**, **b**) and HB (**c**, **d**) models, as a function of maternal age. Gradients of lines correspond to model parameter $\beta$. Datapoints give the mean heteroplasmy change from a sampled set (set of oocytes, or litter) $\pm$ s.e.m. Fit lines are mean and 95% c.i.s for simple linear model fit (c.i.s from bootstrapping with the percentile method) of all individuals within all samples, for each dataset independently. **b**, **d** show a shift between oocyte heteroplasmy dynamics and pup heteroplasmy dynamics

underlying behaviour, we will then jointly infer the values of each parameter in the model using the full datasets, providing a coherent quantitative picture of the processes involved in modulating mtDNA population statistics. We will then demonstrate the power of this stochastic model to make probabilistic predictions of biologically and medically pertinent mtDNA behaviour.

**Time-independent mtDNA heteroplasmy segregation**. We first focus on processes influencing mean heteroplasmy through and between generations (related model parameter: $\delta$). Both pathological[23,24] and non-pathological[25] mtDNA mutations and haplotypes can be selected against between generations, shifting the average heteroplasmy levels between mother and offspring. However, the precise timing of these heteroplasmy shifts is currently unclear, and selection at the oocyte level[23], prenatally in the germline[24], or both[25] have been proposed. These shifts are additionally based on the initial heteroplasmy of the mother, with offspring never[23,24], or only after a restrictive breeding regime[25], reaching 100% of the non-wildtype mtDNA.

First we aim to pinpoint the time-slots at which a heteroplasmy shift can occur during (early) development, thereby characterising parameter $\delta$ (heteroplasmy shift in bottleneck/early development). These heteroplasmy shifts are first analysed time-independently; mouse age will be included subsequently. To assess heteroplasmy shift, we analysed mother-oocyte and mother-pup pairs of two heteroplasmic mouse lines. Figure 3a shows the heteroplasmy levels between mothers and their respective oocytes or pups. In both heteroplasmic mouse models, heteroplasmy levels readily reached 0–100% of all haplotypes (C57BL/6N; LE, HB) without special breeding regimes, which contrasts with existing models[23–25].

In the LE model, the individual heteroplasmy distributions from oocytes and pups are comparable with the corresponding reference measurement (Fig. 3a, b; $p > 0.05$ for Mann–Whitney tests comparing reference and measured heteroplasmies). However, we detected a shift from oocyte to pup heteroplasmy ($p = 0.024$, Mann–Whitney test comparing oocyte and pup heteroplasmies). In contrast, in the HB model, the majority of oocytes have a higher level of HB mtDNA than the respective mother, indicating a biased segregation towards the wild-mouse derived mtDNA ($p \sim 10^{-6}$, Mann–Whitney test as above). HB mice also demonstrated a shift from oocyte to pup heteroplasmies ($p < 10^{-14}$, Mann–Whitney test as above). Interestingly, this shift in heteroplasmy from oocyte to pup is of different directions in the different mouse models, with LE heteroplasmy increasing between generations and HB heteroplasmy decreasing between generations; our full model fit below will characterise this divergence in more detail. Previous work[3] found a difference in segregation directions between LE and HB in some somatic tissues (heart and muscle) that mirror these observations (see Discussion section).

**Time-dependent mtDNA heteroplasmy segregation**. Second, we aimed to see whether heteroplasmy segregation can be influenced by the age of the mother (related model parameter: $\beta$). If a change in mean heteroplasmy does occur over time, there is a possibility of mutations getting lost in consecutive litters of the same mouse. We thereby characterise parameter $\beta$ (segregation between germline and reference tissue over time).

Figure 4 shows mean heteroplasmy behaviour over time, derived from transformed single-cell heteroplasmy measurements in mothers and samples from pups in our two genetically distinct mouse models. We initially view mother and oocyte/pup data as decoupled and explore their behaviours individually (they are used jointly in the umbrella model below). Little support exists for strong changes (Supplementary Note 1); a decrease in pup heteroplasmy with mother's age is observed but is of rather low

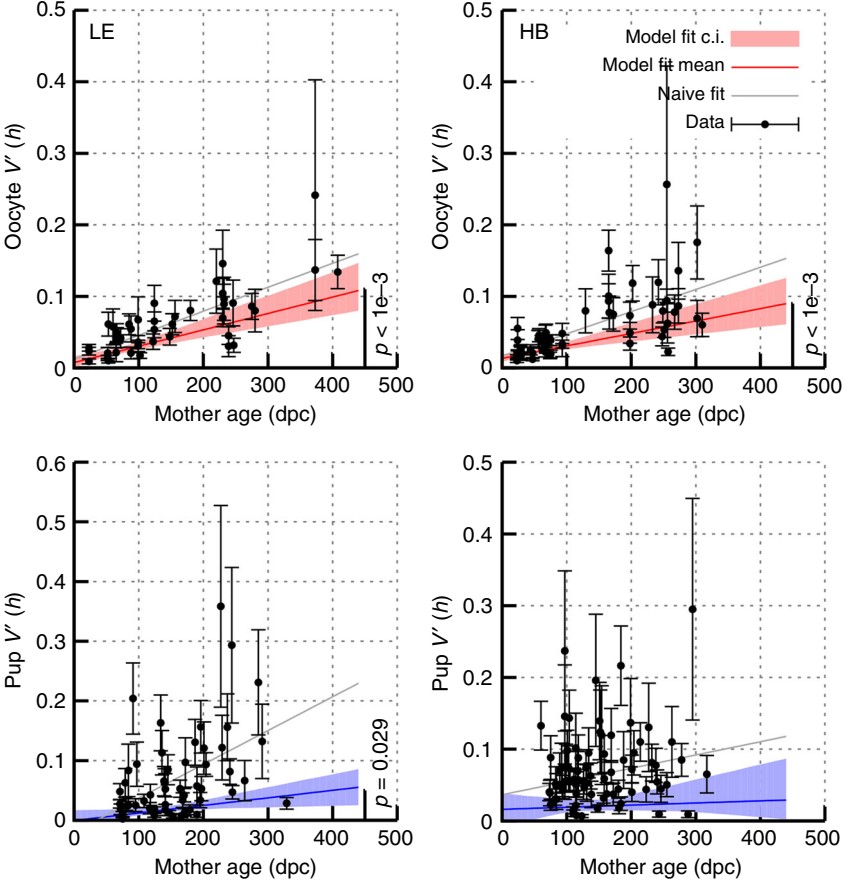

**Fig. 5** Germline heteroplasmy variance increases with organismal age. The cell-to-cell variance in oocyte heteroplasmy increases as a mouse ages; the pup-to-pup variance in litter heteroplasmy increases with the age of the mother. Datapoints give mean normalised heteroplasmy variance with estimated standard errors (see text); fits presented are both a naive linear fit to the mean variance statistics, and a weighted fit (mean and 95% c.i.s) with bootstrapping accounting for variability in the measured heteroplasmy variance statistics (hence penalising the samples with large errorbars)

magnitude – the rate of heteroplasmy change is several times lower than that observed in somatic tissues in previous work[3], suggesting that other processes may play more substantial roles in dictating mtDNA behaviour.

A generational shift in heteroplasmy behaviour with time is visible in both models, with heteroplasmy shifts in pups generally more pronounced than in oocytes: that is, the observed change in heteroplasmy with maternal age is of higher magnitude in pups than in oocytes. This observed difference is confirmed through a likelihood ratio test (Supplementary Note 1, LE, $p = 0.002$; HB, $p < 10^{-16}$), showing statistical support for a mechanism acting to change heteroplasmy between generations, which we characterise with our umbrella model below.

Several other features of the raw data demonstrate results which we will analyse in detail in later sections. Increases in heteroplasmy variance with mother's age can be observed in oocytes and LE pup measurements, manifest as growing error bars over time. The vertical intercept of the mean heteroplasmy measurements is above zero in HB oocytes, reflecting the systematically higher heteroplasmy in oocytes than reference tissue (see above). Justifying one of our modelling assumptions, the distributions of transformed heteroplasmy we observe are not incompatible with being normally distributed (only one of our 232 sample sets displayed $p < 0.001$ for the Kolmogorov-Smirnov normality test; the distribution of $p$-values was roughly uniform, Supplementary Figure 2). We underline here that we are not claiming that heteroplasmy is normally distributed – it is

constrained by the limits zero and one and so cannot generally follow a normal distribution. Rather, our transformation Eq. 4 casts heteroplasmy values onto the full real line, and given the rarity of homoplasmic observations we find the resulting transformed heteroplasmy distributions – which we use in the followup analysis – to be approximately normally distributed.

**Heteroplasmy variance increases linearly over lifetimes**. Having characterised changes in mean heteroplasmy occuring within and between generations, we turn to heteroplasmy variance (related model parameters: $\varphi$ and $\theta$). We aimed to see whether heteroplasmy variance increase with the age of the mother, both in oocytes and pups. We thereby characterize the model parameters $\varphi$ (increasing germline variance with time), and $\theta$ (variance induced in bottleneck/ early development).

To quantify the increase of heteroplasmy variance over time, we first consider the data on oocytes and the data on pups as separate, uncoupled entities. We use the well-known normalisation of $V(h)$ by $\mu(1 - \mu)$, where $\mu$ is mean heteroplasmy, to account for differences between samples, defining $V'(h) = V(h)/\mu(1 - \mu)$. We use the model-free approach using the fourth central moment in ref. 9 to estimate sampling errors in variance measurements. This approach naturally accounts for the difficulty of sampling heteroplasmy variance, and the corresponding uncertainty in variance measurements[9]. We also fit a linear model to the data accounting for the

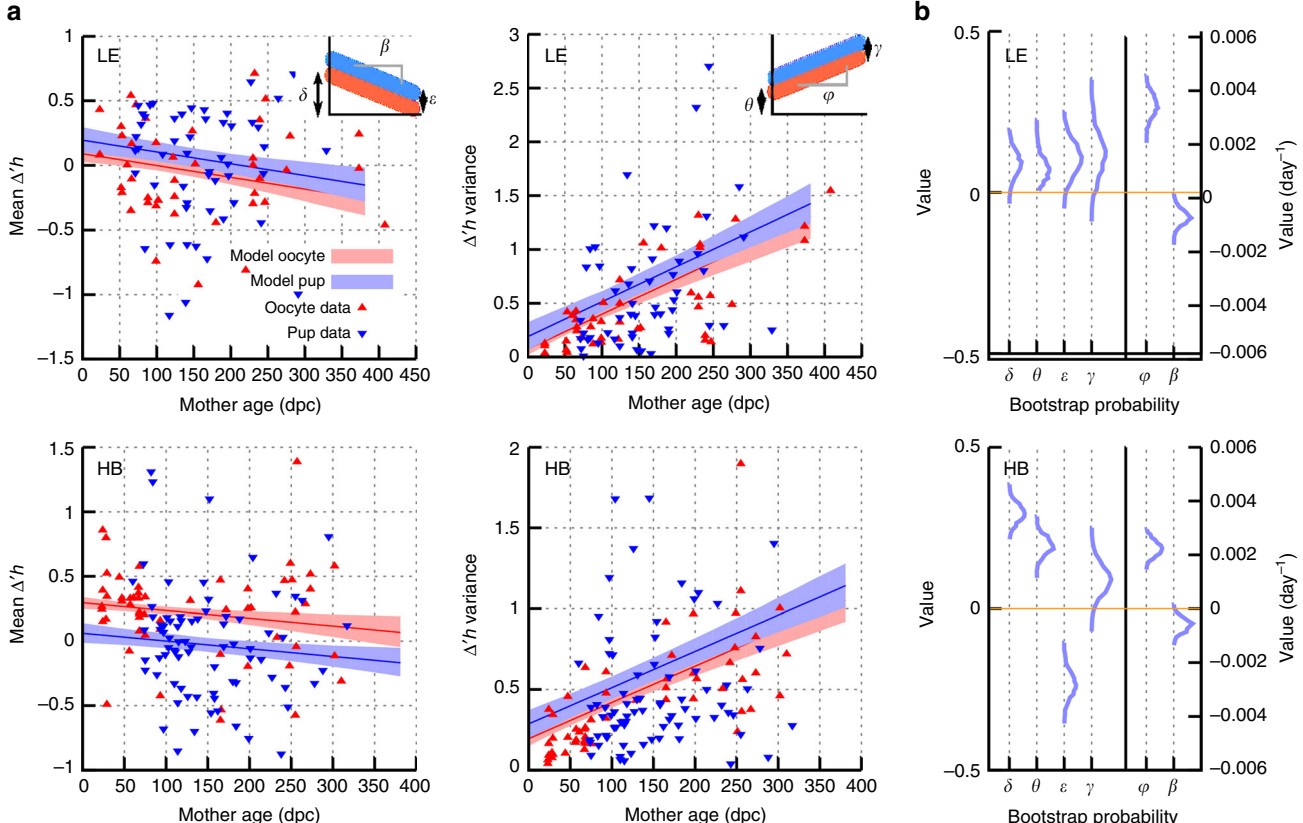

**Fig. 6** Heteroplasmy statistics evolve during ageing, development, and between generations. **a** Joint inference of the model behaviour given the full, combined set of our heteroplasmy observations. An offset decrease in mean heteroplasmy and a linear increase in heteroplasmy variance, suggested by Figs 3 and 4, are clearly observed. Insets show how the inferred behaviour corresponds to the mechanistic parameters in our stochastic model. Shaded regions correspond to 95% c.i.s on summary statistics; the corresponding intervals for individual heteroplasmy measurements are shown in Supplementary Figure 4. **b** Bootstrapped distributions of the parameter values in our model given data from LE and HB mice

uncertainty in variance data, so that points with high uncertainty are correspondingly penalised in the model fit (diminishing, for example, the contribution from the points with large error bars in Fig. 5). Finally, we performed a naive fit to the variance measurements alone with no quantified error, weighting all observations equally. The increase of heteroplasmy variance with time is clear and robust across cases except for the weighted fit in HB pups (Fig. 5; statistical details in Supplementary Note 1). The rate of change of heteroplasmy variance increase of roughly $2 \times 10^{-4}$ day$^{-1}$ is consistent with recent theoretical work[20] and observations in other mouse tissues[3].

We also observe that, for HB oocytes, the inferred behaviour of heteroplasmy variance extrapolates to a nonzero value at fertilisation ($t = 0$), suggesting that an early increase in heteroplasmy variance occurs before the ages involved in our data (hence $\theta \neq 0$). This observation is compatible with processes during development (the mtDNA bottleneck) inducing cell-to-cell variability in the germline, as found in ref. [7], and indeed the observed shift here is of the same magnitude as identified by previous studies in mice[7,22,26], where $V'(h)$ values around 0.02–0.04 have been reported as arising during the developmental bottleneck.

For interpretation, an increase in $V'(h)$ from 0 to 0.1 corresponds, for a mean heteroplasmy of 50%, to an increase from a 0% heteroplasmy interval to a range of 19–81%. Correspondingly, Supplementary Figure 3 shows the maximum heteroplasmy interval in percentage points through time in our dataset, rising, for example, from 22 points to 69 points in LE

oocytes from females aged 3–353 days, and from 13 to 80 points in LE pups from maternal ages 70–244 days.

**Joint inference of all influences on mtDNA statistics**. To harness the statistical power of our combined dataset we now jointly infer the values of each of our mechanistic parameters (Fig. 2a), using two combined datasets, each amalgamating mother and pup data for one of our genetic pairings. We use bootstrapping (see Methods section) to infer distributions for each parameter. We use the percentile method (see Methods section) to determine if these inferred distributions are compatible with a given parameter being zero (and hence the corresponding process not providing an important contribution to mechanisms influencing heteroplasmy).

Figure 6a shows the inferred joint behaviour of mtDNA statistics over time from the combined dataset. Note that the confidence intervals in this plot are for the summary statistics (mean and variance) and are not directly connected to the spread of datapoints: Supplementary Figure 4 shows the corresponding confidence intervals on the spread of heteroplasmy values, illustrating the model's ability to capture heteroplasmy observations. Figure 6b shows the bootstrapped distributions on mechanistic parameters. We find statistical support (all $p$-values and 95% c.i.s from bootstrapping with the percentile method) for nonzero variance increase with time in mothers (nonzero $\varphi$, mean HB = $2.24 \times 10^{-3}$ day$^{-1}$ with $p < 2 \times 10^{-3}$, c.i. (1.83 to 2.71) $\times 10^{-3}$ day$^{-1}$; mean LE = $3.25 \times 10^{-3}$ day$^{-1}$ with $p < 2 \times 10^{-3}$, c.i. (2.58 to 3.87) $\times 10^{-3}$ day$^{-1}$), as expected from

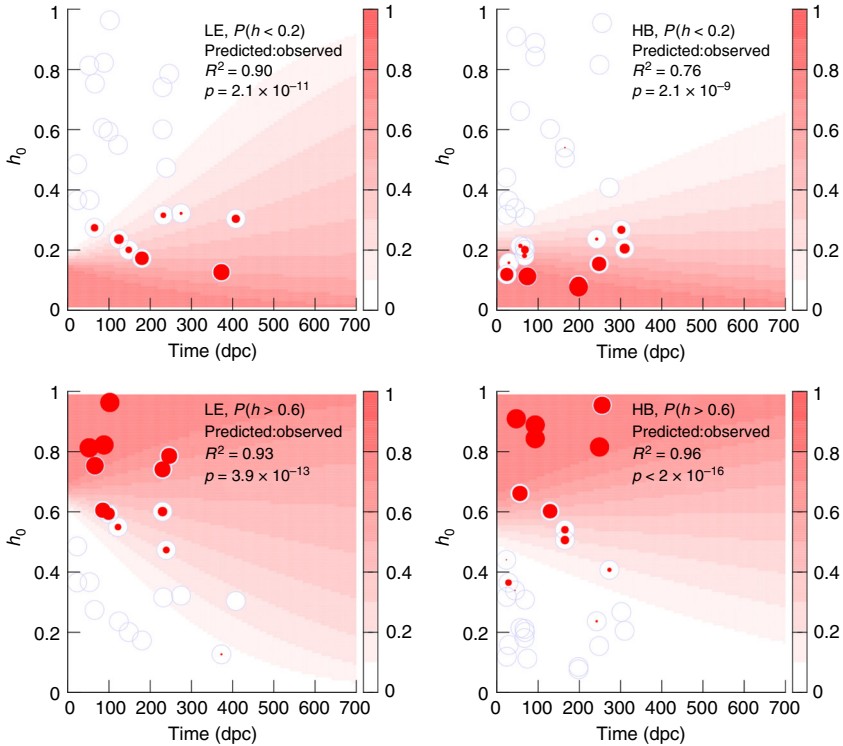

**Fig. 7** Predicting the changing probability of crossing heteroplasmy thresholds over time. Background colour map gives the predicted probability, based on a training subset of data, with which a given heteroplasmy threshold is crossed with time (horizontal axis), for a given initial heteroplasmy (vertical axis). For example, a colour map value of 0.4 at $t = 150$ dpc, $h_0 = 0.5$ means that 40% of cells in a sample with initial heteroplasmy 0.5 are expected to have crossed the threshold after 150 days. Each blue circle corresponds to a sample of oocytes from a separate test subset of our LE and HB datasets, positioned by mother's age and reference heteroplasmy. The radius of red circles within these points gives the proportion of oocytes in that sample that cross the threshold. Agreement between prediction and observation is manifest as agreement between the radius of the red circles and the predicted colour map probabilities. (Top) $P(h < 0.2)$, modelling the probability of obtaining a desired low-heteroplasmy (<20%) embryo from PGD; (bottom) $P(h > 0.6)$, modelling the probability with which cells exceed a (60%) threshold for mitochondrial disease

Fig. 4. We also find support for a weak decrease of mean heteroplasmy throughout mother's lives, corresponding to the mtDNA segregation bias observed in other tissues in these models[3] (nonzero $\beta$, mean HB $= -5.83 \times 10^{-4}$ day$^{-1}$ with $p = 0.008$, c.i. (1.97 to 9.89) $\times 10^{-4}$ day$^{-1}$; mean LE $= -8.89 \times 10^{-4}$ with $p = 0.002$, c.i. ($-4.48$ to $-13.8$) $\times 10^{-4}$ day$^{-1}$). A transient increase in mtDNA variance due to the mtDNA bottleneck is also supported and of magnitude compatible with results from a previous study looking at the bottleneck in detail[7] (nonzero $\theta$, mean HB $= 0.197$ with $p < 2 \times 10^{-3}$, c.i. 0.150 to 0.240; mean LE $= 0.0735$ with $p < 2 \times 10^{-3}$, c.i. 0.0187 to 0.140).

Additionally, we find support for heteroplasmy-based selection of oocytes for the next generation, with the direction of this selection depending on the genetic details of the admixed population (nonzero $\varepsilon$, mean HB $= -0.233$ with $p < 2 \times 10^{-3}$, c.i. $-0.171$ to $-0.300$; mean LE $= 0.105$ with $p = 0.013$, c.i. 0.024 to 0.19). We do not find strong statistical support for nonzero values of the $\gamma$ parameter (mean LE 0.11, c.i. $-0.0079$ to 0.24; mean HB 0.091, c.i. $-0.017$ to 0.168), suggesting that this selection may not strongly modulate heteroplasmy variance. We also note the small magnitudes of the observed decreases of mean heteroplasmy over time; in some somatic tissues we previously found segregation an order of magnitude greater (for example, rates approaching 0.01 day$^{-1}$ in HB muscle).

**Age-dependent probability of heteroplasmy observations**. In addition to verifying theoretical models for mtDNA dynamics[7], the observation of increasing variance has important implications

for fertility strategies (see Discussion section). Motivated by the goal of optimising the implementation of these strategies, our data can be used to parameterise a platform to predict heteroplasmy statistics as organisms age. As transformed heteroplasmy distributions are reasonably normal, we employ a normality assumption to predict the features of heteroplasmy distributions over time, and so the probability of a given cell's heteroplasmy being higher or lower than a given threshold with time.

To compute this probability for a given initial heteroplasmy distribution, we first compute the transformed heteroplasmy change from initial heteroplasmy $h_0$ required to reach a threshold $h$, which is $\Delta' h(h; h_0) = \log((h(h_0 - 1))/(h_0(h - 1)))$ (see Methods section).

We then model the distribution of transformed heteroplasmy as a normal distribution with a linearly increasing variance and with a potentially changing mean. The overall probability of reaching the threshold $h$ is then

$$P(h; h_0, t) = \frac{0.5\,\mathrm{erfc}((\beta t - \Delta(h, h_0)))}{\sqrt{2}\sqrt{(\varphi t + \theta)}}, \quad (3)$$

where erfc$(x)$ is the complementary error function.

After training the model in Eq. 3 on experimental data, it can be used to make predictions about heteroplasmy statistics over time. To illustrate this process, we first trained the model on a training subset of our data (inferring parameters as above, but only using 50% of the data for each haplotype), and then tested its predictions on the

remaining, distinct test subset. Figure 7 shows the success of probabilistic predictions made with this approach, motivated by two medically pertinent cases: the probability that an embryo observed during development (for example, in the context of PGD[4]; see Discussion section) has a heteroplasmy below a given value, and the probability that cells in an ageing organism exceed a given value. It can readily be observed how the probability that a given oocyte will cross a threshold substantially increases over time, validating the predictions of the trained model. To quantify this agreement we analysed the correlation between predicted and observed threshold crossings, with p-values against the null hypothesis of no link between prediction and observation. Each correlation was strong with strong statistical support (Fig. 7).

Implementation of this model to make predictions about optimised PGD requires a connection with human data reporting heteroplasmy data between generations: in the discussion we outline how this platform would be constructed. We also note that the question of when during development to implement a PGD sampling strategy is complex, influenced by the detailed dynamics of the mtDNA bottleneck; previous theory has addressed the question of when best to sample[7].

## Discussion

Both the cell-to-cell mean and the cell-to-cell variance of heteroplasmy are of central importance in the inheritance and onset of mitochondrial diseases, and in the design of therapies to address these diseases. Heteroplasmy mean and variance are related, but not completely coupled, and the cellular pressures governing how they change over time – and indeed, the very questions of whether and when such changes occur – remain hotly debated. Mounting evidence suggests that changes in heteroplasmy mean are influenced by several mechanisms acting in the germ line (purifying selection[23,27]) and/or embryo/foetus (biased segregation[25]); in the pre-implantation embryo[28]; and during gestation[24]. In concert, the mtDNA bottleneck leads to heteroplasmy variance increase in the offspring due to mechanisms that most likely involve a reduction of mtDNA copy number, variance induced through cell divisions, and continued turnover of mtDNA within cells ([7]; reviewed in ref. [29]), but the factors that influence this process are still debated.

This work has explored, in unprecedented detail, the dynamics of these statistics in two mouse models with different genetic distances between their admixed mtDNA populations, and over a wide range of initial heteroplasmy values. We analysed the mtDNA segregation in two heteroplasmic mouse lines, with 1947 oocyte measurements (1152 HB, 795 LE) and 899 pup measurements (552 HB, 347 LE), with mother ages ranging from 24 dpc (i.e. 3 days after birth) to 409 dpc. Among the most important of our results are the observations that (i) germline heteroplasmy variance continues to increase throughout organismal ageing, challenging the picture of a single bottleneck process; (ii) in addition to this time dependence, inferred heteroplasmy variance and segregation statistics may differ if exclusively oocyte, or exclusively pup, samples are analysed; and (iii) the variance increase of heteroplasmy in ageing mothers can be predicted with quantitative theory. These findings are applicable to human reproductive therapies; we outline how future expansion of data characterising the human system will facilitate predictive advances in therapy design.

Our experiments show that heteroplasmy variance increases linearly over time in mammals. This is compatible with theoretical results describing mtDNA turnover[20], our previous results demonstrating linear increase in early post-bottleneck development[7], and results in Drosophila[30–32]. Our observations provide,

to our knowledge, the most detailed quantitative characterisation of single-cell heteroplasmy dynamics in mammals to date. Previous approaches describing mtDNA drift have typically used a coarse-grained heuristic measure of time, often by employing an effective number of generations via the Wright formula (discussed in ref. [20]). Some studies take this coarse-graining to its limit, describing the buildup of heteroplasmy variance as the result of a single effective bottleneck event which combines the developmental bottleneck and the subsequent buildup of drift. Without a detailed quantitative accounting for drift, this simplification leads to heterogeneity in the reported bottleneck size[7,26,33,34]. Our results simultaneously illustrate the importance of, and provide quantitative means to allow, accounting explicitly for separated drift and developmental processes for determing heteroplasmy distributions. For this reason, we advocate using the term "heteroplasmy variance" rather than the shorthand "bottleneck", to explicitly separate out contributions to variance from the developmental bottleneck and from ongoing drift.

The genetic bottleneck can account for random (but marked) changes in mtDNA heteroplasmy between generations[7]. However, it is becoming increasingly evident that additional mechanisms influence heteroplasmy dynamics between generations[11]. The increase in variance due to the bottleneck is sequence-dependent[17], and evidence exists that both pathological and apparently non-pathological mtDNA mutations can be selected against between generations (recently reviewed in ref. [10]). Mice expressing a proofreading-deficient mitochondrial DNA polymerase, non-synonymous mutations were underrepresented compared to synonymous mutations in the mtDNA of the offspring[27]. In another study a severe heteroplasmic mt-Nd6 mutation was eliminated within four generations[23]. This selection seems to act intraovarially, most likely by removal of affected oocytes. In contrast, a slightly deleterious tRNA[Met] mutation was selected against in the developing embryo, by selection acting at the cell or organelle level in the embryo[24].

In addition to these pathological cases, a heteroplasmic mouse model harbouring 129S6 and NZB mtDNA (both apparently non-pathological haplotypes), heteroplasmy segregated to undetectable levels of NZB mtDNA within two generations. In contrast, it took ten generations of selective breeding to reach a single female that proved to be 100% homoplasmic for NZB. In this case selection seemed to work partly in the ovary, and partly during gestation[25]. Taken together, these observations suggest that both pathological and non-pathological mtDNA pairings experience segregation between generations[10].

Here we show, with a much larger dataset, the dynamics of two different segregation regimes (LE, more neutral; HB, more biased) through ageing and between generations. Both LE and HB mtDNA haplotypes derive from wild-derived mouse strains, and both heteroplasmic lines reach 100% homoplasmy without selective breeding. This capacity to reach homoplasmy sets our lines apart from previous pathogenic and non-pathogenic models, apparently avoiding issues from mixing mitochondria[35] and allows a simpler disambiguation of segregation bias and variance increase in mtDNA dynamics.

We prove that segregation bias can already occur at the oocyte level (HB oocytes, Fig. 3). This shift appears less pronounced when observing samples from pups in the next generation, suggesting that processes may exist to ameliorate this bias between generations. While we already showed in a recent report that heteroplasmy levels can change during gestation in a tissue-specific way (HB, heart[3]), the changes found in this study are best explained by selection of cells with relatively lower heteroplasmy levels in the germ line early in development, as compared to the somatic precursor cells. Based on analysis of two heteroplasmic rhesus monkey fetuses a "preimplantation bottleneck" was

proposed[28]. Further research is necessary to see if a similar mechanism works both in rhesus monkeys and mouse, but the timing seems to be comparable.

Our study shows that when analysing offspring or oocyte data separately, heteroplasmy variance statistics may differ (Fig. 4). Correspondingly, heteroplasmy shifts between mother and offspring may not reflect those between mother and oocyte due to shifts during oogenesis or embryonic development (Fig. 3). Taken together, these results clearly indicate that, for a complete picture of mtDNA segregation (and of the bottleneck in particular), both oocyte and offspring data, together with the age of the mother at the time of analysis/ birth should ideally be available (Fig. 6). An umbrella picture where the developmental history of, and relationship between, measured samples increases the power with which mechanisms can be identified, as demonstrated in recent work on mtDNA segregation in mice[3], and in mtDNA dynamics in humans[8], which notably employed an "ontogenic phylogeny" for developmental history to jointly infer dynamics from observations from ref. [36].

Increasing variance in oocyte heteroplasmy is desirable for therapeutic approaches like PGD, where a greater variance of heteroplasmies increases the probability that a low-heteroplasmy embryo can be identified[4]. For a prospective mother carrying a pathological mtDNA mutation at some heteroplasmy, therefore, a rational approach (based on mtDNA statistics alone) would be to undergo PGD at later ages, where the probability of obtaining a low-heteroplasmy embryo is higher. Of course this must be tempered by possible negative effects of increased age in fertility.

The model in Eq. 3 and illustrated in Fig. 7 provides a mechanism to leverage our findings on increasing heteroplasmy variance to optimise therapeutic strategies. To implement this prediction of threshold crossing probabilities and timescales in a clinical context, a (potentially small) set of training data would be required, providing a parameterisation for Eq. 3. Appropriate data would consist of coupled samples of mean mother heteroplasmy and samples of heteroplasmy in offspring produced at different maternal ages. Examples of this type of data do exist in the literature[36] but often amalgamate heteroplasmy levels of different mtDNA mutations, challenging our ability to control for the potentially different levels of selection acting upon each mutation. Data focussed on a small number of mtDNA variants, in conjunction with Eq 3, would characterise the increase in heteroplasmy variance over time in the human system (which importantly may differ according to the genetic particulars of the system[37], allowing predictions of threshold crossing and other statistics of importance in fertility treatments in the same manner as illustrated in Fig. 7. The probabilistic benefit arising from later fertility treatment can then be weighed against the probabilistic risk of an age-related fertility issue, and the costs and benefits of older PGD therapies can be quantitatively assessed.

## Methods

**Two heteroplasmic mouse models with mtDNA admixtures.** The study was discussed and approved by the institutional ethics committee in accordance with Good Scientific Practice (GSP) guidelines and national legislation. FELASA recommendations for the health monitoring of SPF mice were followed.

Heteroplasmic mice were obtained from two heteroplasmic mouse lines (denoted HB (Hohenberg) and LE (Lehsten), according to the German localities of original collection) we created previously by ooplasmic transfer[3]. These mouse lines contain the nuclear DNA of the C57BL/6N mouse, and mtDNAs both of C57BL/6N and of a wild-derived house mouse (either HB or LE). All three mtDNA variants belong to the same subspecies, *Mus musculus domesticus*. For details on sequence divergence see[3].

**Ear-clip and tail reference tissue and DNA extraction.** All mother and offspring calculations are measured in either ear-clip or tail samples that were obtained at the weaning (at the age of 21 days) of the respective animals[24]. Samples were stored at

−20 °C. DNA was extracted using the NucleoSpin Tissue Kit (Macherey-Nagel, Germany) according to the protocol for animal tissue (no RNase treatment).

**Isolation and lysis of oocytes.** Mice were killed at the indicated ages by cervical dislocation. Ovaries were extracted and immediately placed in cryo-buffer containing 50% PBS, 25% ethylene glycol and 25% DMSO (Sigma-Aldrich, Austria) and stored at −80 °C.

For oocyte extraction, ovaries were placed into a drop of cryo-buffer and disrupted using scalpel and forceps. Oocytes were collected and remaining cumulus cells were removed mechanically by repeated careful suction through glass capillaries. Naked oocytes were then washed in PBS before they were individually placed into compartments of 96-well PCR plates (Life Technologies, Austria) containing 10 μl oocyte-lysis buffer[28] composed of 50 mM Tris-HCl, (pH 8.5), 1 mM EDTA, 0.5% Tween-20 (Sigma-Aldrich) and 200 μg/ml Proteinase-K (Macherey-Nagel). Samples covered stages from primary oocytes of 3-day-old mice up to major oocytes of adult mice. Samples were lysed at 55 °C for 2 h, and incubated at 95 °C for 10 min to inactivate Proteinase K. The cellular DNA was finally diluted in 190 μl Tris-EDTA buffer, pH 8.0 (Sigma-Aldrich).

**Heteroplasmy quantification by ARMS-qPCR.** Heteroplasmy quantification was performed by Amplification Refractory Mutation System (ARMS)-qPCR[3,38–40].

Consensus assay (103 bp amplicon): Co2-f: TCTTATATGGCCTACCCATTCC AA, Co2-r: GGAAAACAATTATTAGTGTGTGATCATG, Co2-FAM: FAM-TTG GTCTACAAGACGCCACATCCCCT-BHQ1

ARMS assays:

*1. mt-Rnr2* assay (for HB, LE and C57BL/6N; 142 bp amplicon)

16SrRNA2340(3)G-f: AAATCAACATATCTTATTGACCgAG, where the small letter designates a base mismatching with both the target and the non-target alleles (haplotype C57BL/6N; used for heteroplasmy analysis of LE samples)

16SrRNA2340(3)A-f: AATCAACATATCTTATTGACCgAA (haplotype HB, LE),

16SrRNA2458-r: CAC CAT TGG GAT GTC CTG ATC, 16SrRNA-FAM: FAM-CAA TTA GGG TTT ACG ACC TCG ATG TT-BHQ1

*2. mt-Cyb* assay (for C57BL/6N; used for HB heteroplasmy analysis; 79 bp amplicon)

Cyb240(3)-f C57 240 ARMS: TAGCAATCGTTCACCTCgTC, Cyb318-r: ATT TTATCTGCATCTGAGTTTAAT, Cyb-FAM: FAM-ACGAAACAGGATCAAAC AACCCAACAGG-BHQ1

Every qPCR run included the consensus and the ARMS assay each performed in triplicate. The master mix contained 1 × buffer B2 (Solis BioDyne, Estonia), 4.5 mM MgCl2, 200 μM of dATP, dCTP, dGTP, and dTTP (dNTPs, Solis BioDyne), 300 nM of each primer, 100 nM hydrolysis probe (Sigma-Aldrich) and HOT FIREPol DNA polymerase according to the manufacturer's instructions (Solis BioDyne). Per reaction 12 μl master mix and 3 μl DNA were transferred to a 384-well PCR plate (4titude Ltd, United Kingdom) using the automated pipetting system epMotion 5075TMX (Eppendorf, Germany). Amplification was performed on the ViiA 7 Real-Time PCR System operated by the ViiA™ 7 Software v1.1 (Life Technologies, USA). DNA denaturation and enzyme activation were performed for 15 min at 95 °C. DNA was amplified over 40 cycles consisting of 95 °C for 20 s, 58 °C for 20 s and 72 °C for 40 s.

The standard curve method was applied for determination of qPCR efficiency. Briefly, amplification efficiencies calculated in each run separately from a dilution series of mouse DNA harbouring 100% of the respective mitochondrial haplotype ranged from 0.87 to 0.94 ($R^2 > 0.99$; $y$ intercepts from 28.3 to 34.5). All experimental samples were covered by the linear dynamic range of the standard curve. To assess assay specificity, each run contained a mouse DNA harbouring the non-target mtDNA type of the respective heteroplasmic combination (i.e., C57BL/6N or HB mtDNA). All assays discriminated C57BL/6N from HB or LE mtDNA with a sensitivity of at least 0.5%. Absence of inhibition in a sample DNA was regularly confirmed based on $Cq$ value obtained from diluting the sample in Tris-EDTA buffer (pH 8.0, Sigma-Aldrich).

Mitochondrial heteroplasmy was always calculated from the assay detecting the minor allele (C57BL/6N or wild-derived (HB or LE) <50%). If both specific assays gave values of more or around 50%, the mean value of both assays was taken. Each qPCR run contained a mandatory no template control (NTC) for each assay ($Cq > 40$). The study was conducted according to the minimum information for publication of quantitative real-time PCR experiments[3,41].

**Transformation and analysis of heteroplasmy statistics.** All analyses in this article are platform-independent and the statistical procedures corresponding to each section are described below.

Being a population fraction, heteroplasmy is bounded by zero and one and is expected to vary nonlinearly with time even when the dynamics of individual mtDNA species change linearly. The same selective pressure inducing a heteroplasmy shift will lead to different absolute heteroplasmy changes for different initial heteroplasmies. This nonlinearity also affects the structure of cell-to-cell heteroplasmy distributions, which can be highly non-normal[9,16], limiting the interpretation of variance statistics. A well-known method to avoid this is to normalise heteroplasmy variance by $\mu(1 − \mu)$, where $\mu$ is mean heteroplasmy.

To overcome these difficulties, we use transformed heteroplasmy, which expresses a change in heteroplasmy, from a reference value $h_0$ to a sampled value $h$, as the selective difference required to induce that change. We will use an early heteroplasmy measurement in a reference tissue in a mother (tail or ear) for $h_0$; $h$ corresponds to a later measurement in an oocyte or pup. Specifically

$$\Delta' h(h; h_0) = \ln\left(\frac{h(h_0 - 1)}{h_0(h - 1)}\right) \qquad (4)$$

This transformation controls for initial heteroplasmy and so facilitates the comparison of heterogeneous individuals in large datasets[3,20]. It is motivated by a mathematical description of stochastic mtDNA dynamics[20] and recapitulates existing methods based on fitness differences[22]. A weakness of this transformation is that it cannot, without correction, address homoplasmic cells, which lead to divergences in Eq. 4, but such cells make up a very small fraction of our dataset and do not substantially influence our results.

We will invoke a normality assumption for cell-to-cell distributions of transformed heteroplasmy. That transformed heteroplasmy values should approximate a normal distribution can readily be seen by applying the transformation in Eq. 4 to theoretical distributions of heteroplasmy[13], and we demonstrate below that this assumption is compatible with our data. In practise, measurement errors in heteroplasmy sampling can exist; in this model, these errors can confound the inference of initial changes in variance ($\theta$ in our subsequent model), but unless systematic changes in measurement error with organismal age exist, inferences about the time behaviour of heteroplasmy statistics ($\beta$ and $\varphi$ in our subsequent model) will be robust with respect to these errors.

**Data availability**. All raw heteroplasmy data are available as Supplementary Data 1. Illustrative Mathematica notebooks containing the full analysis of heteroplasmy statistics are available upon request.

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

## Acknowledgements

J.P. acknowledges support from CSF grant 16-23773S. I.G.J. acknowledges support from the University of Birmingham via a Birmingham Fellowship. N.S.J. acknowledges grant support from the BHF (RE/13/2/30182) and EPSRC (EP/N014529/1).

## Author contributions

J.P.B., G.B., N.S.J., I.G.J., T.R., T.K. conceived the study; T.K., T.R., J.P. produced and maintained the mouse models; J.P.B., T.K., V.H., T.R., S.H. performed the experiments; N.S.J. and I.G.J. developed the modelling and statistical machinery; I.G.J. performed the mathematical analysis; J.P.B., N.S.J., I.G.J., J.P., R.S. interpreted and analysed the results; I.G.J., J.P.B., N.S.J., wrote the paper; all authors contributed to editing the paper.

## Additional information

**Competing interests:** The authors declare no competing interests.

