## [Peer Review File · Nature Communications]

Reviewers' comments:

Reviewer #1, an expert in mitochondrial genetics (Remarks to the Author):

This is an impressive and useful experiment that has developed a multigenerational mouse model of heteroplasmy inheritance. In general the mathematics is handled correctly, but I have several serious problems with how the data is interpreted and presented.

Major concerns

1: My most important concern is about a statement that the authors make repeatedly throughout this paper, starting in the abstract, highlight number 1, the introduction (line 95), and throughout the Discussion. The "single bottleneck picture" as presented by the authors is a strawman. The standard paradigm in this field for about 20 years now has been "bottleneck + drift" which is completely consistent with the observations reported here. There is a heteroplasmy shift between mother and offspring during embryogenesis (the bottleneck) and then continued intracellular heteroplasmy changes over time as mtDNA molecules are replicated and destroyed (drift). I absolutely would not read interpret the data in this paper as "implying that "bottleneck size" changes with age", as the authors repeatedly state. It just means that drift proceeds as age increases. Remarkably, the word "drift" only appears in this paper in one of the cited paper names. The authors avoid using the standard terminology in this field (drift) for one of the major quantities that they are measuring, reporting and discussing. Repeatedly they talk about this feature as if they are the first to discover it, when actually drift is a traditional concept in the field.

2: My second major concern is about the great leap between this mouse model and human application. The authors often overstate the case here. For example, the final sentence in the Introduction (lines 103-104) "Our findings have direct implications for our understanding of mtDNA inheritance and disease manifestation, and thus for human reproductive techniques" is worded too strongly. The difference between a few days drift in a mouse model and several decades of drift in a human are enormous, making it hard to "directly" apply results from this animal model to humans. This problem is actually made worse when we get into the Results section (discussed below) and find that the results from the two mouse models are inconsistent with each other. The wording on extending the results of these mouse models to lessons in human reproduction is inconsistent throughout the paper. Sometimes it is appropriately cautious (lines 372-374, 504-516), while at other times it is overselling the applicability (lines 102-104, 414-416).

3: Lines 147-148: To have analyses available only in a proprietary format (Mathematica) is a barrier to reproducibility. The analyses should be reported in either a free and open source format (such as R) or better yet, simple reported as plain text that is readable by all.

4: Lines 244-246: The authors mention in results that their mouse model reaches 0% to 100% in all haplotypes, in contrast to three other existing mouse models of heteroplasmy. As far as I see, they do not follow up on why this difference occurs in the discussion. This major difference from the other established mouse models is certainly worth some discussion.

5: Line 249: I would disagree with the wording "This suggests that mtDNA segregation in this mouse model is of low enough magnitude to appear neutral.". Based on the width of the distribution of heteroplasmies visible in Fig 3A the amount of segregation is pretty comparable in the LE and HB models. The difference is that the LE model shows much smaller bias in heteroplasmy shift. But based on the data I can see here, I would not call that due to low magnitude of segregation. The segregation magnitude appears quite high in both models. Since the authors do not give a definition of what they mean by "magnitude" of "mtDNA segregation", it

is hard to sort this out.

6: The shift in LE oocytes and in LE pups is small but from the Fig 3B graph appears to be statistically significant (unless there has been a mistake in the error bar definition. See point 8 below.) No stats are reported for this, and the text description is just that they values are "comparable", which is open to interpretation. More detail is warranted here.

7: Figure 3. p values are given with no explanation of what is being tested or which statistical test is done. More detail must be given. These tests are complicated by the non-normal distributions of the heteroplasmy values, as they are strictly bounded by 0 and 1, so it is important that the reader know what was done here to calculate those p-values. While I believe the spirit of these p-values is correct just by viewing the data plots in Fig 3A, the details do matter and should be given.

8: Fig 3B The Y axis says "mean and s.e.m.". Standard practice would be to use 2 x SEM for error bars, to approximate 95% confidence intervals. It is not clear to me whether this is an oversight in the axes label, or a mistake in the presentation of the error bar sizes.

9: Fig 3: The statistically significant difference between the heteroplasmy shift in the oocytes compared to the shift in the pups is rather important. Both the LE and HB models show significant differences, but in opposite directions. The meaning of this shift, and the difference in the shift between the two mouse models needs to be dealt with more fully than it is now. This is the most important result in the paper, and I am just left with the message that sometimes one thing happens and sometimes the opposite happens. Hardly enlightening.

10: Line 287-290: This data on the distribution of p-values should be reported, at least in a supplementary figure.

11: Line 309-314: The authors discuss the extrapolated heteroplasmy variance at time 0, but do not actually report the value and confidence interval of that extrapolated number. The authors state that "the observed shift is of the same magnitude as identified by previous studies..." but the reader cannot judge that since the magnitude is never reported here.

12: Line 324-343: Model parameters are reported here with p values but with no confidence intervals. Please add confidence intervals throughout. The model parameters are shown in the plot and are reported throughout the text here. It would be convenient for the reader if these complete model parameters, along with confidence intervals, were also given in a simple supplementary table.

13: Line 339 "We do not find support for non-zero lambda". Based on Fig 6B, this looks borderline to me. Numerical values with confidence intervals and p values should really be given (either in text or a table) to support this statement.

14: Line 362 "and so set beta =0". This is incompatible with your data from both the HB and LE models which show significantly nonzero beta values. Why do a model calculation that is clearly inconsistent with the data presented in this paper?

15: Lines 345-374 "Probability of observing heteroplasmy above or below a given value with age." This section is highly problematic. The interpretation given here is described as "in the context of preimplantation genetic diagnosis". The data presented in this paper shows strongly significant differences between the two mouse models, meaning that the parameter values derived from this data are certainly not universal for mammals, or even for mice.

16: Lines 376-390 "Evidence for increasing heteroplasmy variance in somatic tissues." This short section is tacked on and does not really fit with the rest of the paper. The data here is from tissue

samples, not single cell samples, and heteroplasmy variances measured at the single cell level and at the multi-cell level will be quite different just from basic math. Single cell variances will be much higher than multi-cell variances. The data is only partially given in Fig S2 and data from skin and spleen samples is mentioned in text but never presented to the reader. Either this section should be presented in full with complete data or it should be dropped. I suggest it be dropped as not truly relevant to the paper.

17: Lines 478-480 "In this study we show that heteroplasmy variance increases continuously through time in organismal ageing. The traditional interpretation of this result would be that "bottleneck size" decreases with age..." No. The traditional interpretation would be that this is drift.

18: The fundamental result of this work is that mother-oocyte and mother-pup heteroplasmy shifts disagree and are even in statistically significantly opposite directions. Furthermore, the direction of that difference is opposite in the two mouse models. This fundamental result is not dealt with in sufficient detail in this paper. Possible reasons for these differences need to be discussed in more detail. It is particularly confusing to me as a reader that in Figure S1, which is supposed to give the reader a visual explanation of the possible dynamics of this system, presents 4 possibilities yet none of these correspond to the actual data in this paper showing opposite direction in mean in oocytes and pups. All four possibilities presented show the oocytes and pups behaving in exactly the same manner.

Minor points

19: This may be a minor point, but I found it quite distracting that the origin of the important label "LE" and "HB" was never given in the paper. As a reader, it left me wondering whether I had missed some important information in the meaning of these labels, or whether they labels were arbitrary. A simple statement of that at the first use of the LE and HB terms in the Methods would greatly improve the readability of this manuscript.

20: Please add "days" throughout when reporting mouse ages. Often the authors use "...mice at age 21." or similar phrases without units.

21: Line 450: (HB oocytes, Fig 2)". I believe they mean Fig 3.

Reviewer #2, an expert in statistical modelling (Remarks to the Author):

This is an interesting paper. However, the paper could be improved by considering the following specific points.

1. The abstract and highlights could be improved by including numerical examples.
2. The authors should explain exactly how the new findings might lead to improvements in understanding and prevention of genetic diseases. How might the findings be 'translated'? How could they have general applicability? Much more explanation is needed.
3. The statistical developments seem appropriate, but could be improved by much more use of numerical examples throughout.

Reviewer #3, an expert in modelling (Remarks to the Author):

In this manuscript, the authors presented a study on the dynamics of mtDNA heteroplasmy in mice (two heteroplasmic mice HB and LE). Tissue samples from mothers (ear clip/tail) were taken

at 21 day of age, and used as reference for quantifying changes in heteroplasmy in oocytes at different maternal age and in pups. The authors proposed a simple model of mtDNA heteroplasmy dynamics based on the assumption that mtDNA heteroplasmy follows a Gaussian distribution. The authors found that mtDNA heteroplasmy displays increased variance with age in germline and pups, as well as in somatic tissues. They also observed a mean shift in heteroplasmy between generations. Meanwhile, the heteroplasmy in germline appeared to diverge from the somatic precursors early in development.

Comments:

1. I found the study to be very timely and the data presented were impressive. The observations and conclusions were supported by the data, if the model can be trusted (see below). The idea that mtDNA bottleneck sizes vary with age appears to suggest that the mtDNA bottleneck occurs postnatally. The timing of the bottleneck has indeed been under much debate (for example see Wai et al., Nat. Genet., 40, 1484–1488, 2008). A large body of studies, including a very recent paper from Chinnery lab (Floros et al, Nat. Cell Biol., 20, 144–151, 2018) put the bottleneck event during germline development. I hope the authors would expand the analysis of their dataset in the context of the timing of the bottleneck.

2. The theoretical development of the model used in the analysis of the dataset needs more detail. As a mathematically-inclined person, I hope to see more consistent mathematical notation used in the manuscript and clear definition of the variables. For example, Δh was defined in M1 without arguments, but later in Eq. 1 and 2, have arguments that were never defined. Later on (in the paragraph before Eqn 3), Δh was described as Δh with two arguments (h, h_0). In addition, Eq. (3) has a square root function, but I could not be sure if the square root was for all terms in the bracket. I recommend the authors to use an equation editor to write their equations (e.g. Word Equation). In addition, Figure 2B is inconsistent with Eq. 1 and 2. The y-axis of Fig. 2B shows $h-h_{ref}$. Am I right to assume that the Gaussian distribution describes the difference in heteroplasmy h measured and a reference heteroplasmy h_{ref} . If so, then it is very hard to accept that Δh should be Gaussian.

3. I hope to see more support that the Gaussian distribution assumption is justifiable for their dataset. The non-rejection of Kolmogorov-Smirnov test needs to be supported by other tests of normalcy, e.g. using normal plot.

4. Looking at Figure 6 A alone, I am not entirely convinced that the model as proposed in the study, could explain the data. What were the p-values of the joint inference? How much was the variability in the data that could be explained by the model? Am I right to interpret the shaded area as 95% confidence interval? If so, a large number of data points are outside this region. Furthermore, looking at Figure 6 B, I can see that there is a possible parameter identifiability issue when fitting LE dataset, where the probability of having parameters of different signs are not insignificant. The parameter inference for the HB dataset appeared better. While I am a proponent of keeping any model simple, I am not convinced that the model used in this study is able to explain the data.

Reviewer #4, an expert in mitochondrial heteroplasmy (Remarks to the Author):

The authors present a large interesting dataset, and interpret this in the context of novel mathematical models. The main conclusion - of a maternal age effect on the level of heteroplasmy in the pups - has not been demonstrated before. It is, however, a very weak effect.

I agree that we should refer to heteroplasmy variance not bottleneck. However, I think the authors are adding to the confusion by mentioning a 'changing bottleneck', especially in the abstract. I appreciate that the authors are mathematicians, and that the bottleneck can be considered a

theoretical construct leading to varied heteroplasmy levels. However, most biologists think in material terms, not the abstract. To them, the bottleneck is a biological process, so the 'bottleneck size' does not change with age. If I understand it correctly, the authors are really saying that the computed bottleneck size differs over time because the variance in heteroplasmy increases with age. This may be true – but the changing parameter here is the mathematical concept of bottleneck size, not necessarily the actual bottleneck itself, which has a biological basis. I think it would help to clarify this throughout the manuscript.

How do the authors account for the extreme outliers? In Fig 5, particularly the upper 2 panels, one of two data points have a very wide confidence interval. Why is this, and are these points influencing the weak signal they measure over time? I would like to see the analysis repeated without these unusual data points, to see if the reported relationships are robust.

Related to this, can the authors comment on the number of individual heteroplasmy measurements that are required to compute heteroplasmy variants. Samuels has done some work on this. I presume that there is a relationship between the confidence intervals and the number of measurements in each mouse/litter. Can they show that reduced numbers of measurements (eg in an ageing mouse, with lower fecundity) do not account for their observations?

To my mind there is an inconsistency in the logic that needs to be addressed. If there is selection – as they observed – how can the distributions approximate a normal distribution? Surely it is skewed (ie with the top sliced off). How does this affect their results?

A major finding is that different tissues behave differently over time. Can the authors show that their choice of sample (ear / tail biopsies) are not the 'cause' of the problem?

Related to this, are biopsies done on the same day yielding the same result?

It is disappointing that in the discussion they say that appropriate human datasets exist in the literature (line 508), yet they have not gone on to apply their formulae to this data - this would seem like an obvious and straightforward conclusion to the paper.

I also think that they must be more cautious in their predictions for humans. Some mutations eg m.3243 behave in a very unusual fashion, with the loss of mutation in blood. This demonstrates proof of principle that different mutations behave differently in different tissues.

Minor:

line 339 – what is 'non zero y'?

It should be clearer that the heteroplasmy variance data is indeed single-cell data in the oocytes, but the pup heteroplasmy variance is only measured on a pup-to-pup level, and not in single cells. In some cases they have made this distinction clear (e.g. in the figure 3 legend), but at other points it is implied that the pup heteroplasmy data is single-cell (e.g. line 264).

It would be nice to see a breakdown of the n-numbers in the supplementary data - i.e. 795 LE oocyte samples - how many females, age at harvesting for each animal, how many oocytes from each etc. Same for the pups. Perhaps this will already be submitted with the final paper

We are very grateful to all the reviewers for their time spent with our manuscript and for their insightful and positive comments. Several important points were raised; we have addressed these point-by-point below, with changes highlighted in the revised manuscript. We believe that our work has been improved and strengthened as a result and reiterate our gratitude to these experts who have helped in this process.

Reviewers' comments:

Reviewer #1, an expert in mitochondrial genetics (Remarks to the Author):

This is an impressive and useful experiment that has developed a multigenerational mouse model of heteroplasmy inheritance. In general the mathematics is handled correctly, but I have several serious problems with how the data is interpreted and presented.

Major concerns

1: My most important concern is about a statement that the authors make repeatedly throughout this paper, starting in the abstract, highlight number 1, the introduction (line 95), and throughout the Discussion. The "single bottleneck picture" as presented by the authors is a strawman. The standard paradigm in this field for about 20 years now has been "bottleneck + drift" which is completely consistent with the observations reported here. There is a heteroplasmy shift between mother and offspring during embryogenesis (the bottleneck) and then continued intracellular heteroplasmy changes over time as mtDNA molecules are replicated and destroyed (drift). I absolutely would not read interpret the data in this paper as "implying that "bottleneck size" changes with age", as the authors repeatedly state. It just means that drift proceeds as age increases. Remarkably, the word "drift" only appears in this paper in one of the cited paper names. The authors avoid using the standard terminology in this field (drift) for one of the major quantities that they are measuring, reporting and discussing. Repeatedly they talk about this feature as if they are the first to discover it, when actually drift is a traditional concept in the field.

We're grateful to the reviewer for this comment on the contextualisation of our research. We have included several updates in response. First, we raise the concept of drift in the introduction with several corresponding references, and underline that it is our detailed quantitative characterisation of this process that is a key result of our paper. We have replaced the discussion point about reinterpreting the bottleneck with a description of the difficulty of disambiguating the bottleneck and ongoing drift, and described the value of our research in contributing to this disambiguation. It is our experience that a substantial amount of ongoing research does conflate drift with other sources of variance increase, and we do believe that a discussion of this point, coupled with our detailed characterisation of variance dynamics, is a relevant and important contribution to the literature: we hope the reviewer now finds the ms appropriately contextualised.

2: My second major concern is about the great leap between this mouse model and human application. The authors often overstate the case here. For example, the final sentence in the Introduction (lines 103-104) "Our findings have direct implications for our understanding of mtDNA inheritance and disease manifestation, and thus for human reproductive techniques" is worded too strongly. The difference between a few days drift in a mouse model and several decades of drift in a human are enormous, making it hard to "directly" apply results from this animal model to humans. This problem is actually made

worse when we get into the Results section (discussed below) and find that the results from the two mouse models are inconsistent with each other. The wording on extending the results of these mouse models to lessons in human reproduction is inconsistent throughout the paper. Sometimes it is appropriately cautious (lines 372-374, 504-516), while at other times it is overselling the applicability (lines 102-104, 414-416).

We have removed the inappropriately strong phrasing of “direct” application. In addition, in response to other reviewer comments, we have included extra results illustrating the application of our modelling process to describe and predict heteroplasmy dynamics in the human context using recently available data.

3: Lines 147-148: To have analyses available only in a proprietary format (Mathematica) is a barrier to reproducibility. The analyses should be reported in either a free and open source format (such as R) or better yet, simple reported as plain text that is readable by all.

We did intend to report our analyses as plain text in addition to providing the Mathematica notebook for convenience, and apologise that our methodological description was insufficiently clear. We have expanded the description and made it clear that proprietary software is not required for our analysis.

4: Lines 244-246: The authors mention in results that their mouse model reaches 0% to 100% in all haplotypes, in contrast to three other existing mouse models of heteroplasmy. As far as I see, they do not follow up on why this difference occurs in the discussion. This major difference from the other established mouse models is certainly worth some discussion.

We have expanded our discussion of our models in the context of other pathological and non-pathological mtDNA pairings. Without detailed physiological measurements, we are cautious about speculating on the reasons for this difference, but hope to address this question in future research.

5: Line 249: I would disagree with the wording "This suggests that mtDNA segregation in this mouse model is of low enough magnitude to appear neutral.". Based on the width of the distribution of heteroplasmies visible in Fig 3A the amount of segregation is pretty comparable in the LE and HB models. The difference is that the LE model shows much smaller bias in heteroplasmy shift. But based on the data I can see here, I would not call that due to low magnitude of segregation. The segregation magnitude appears quite high in both models. Since the authors do not give a definition of what they mean by “magnitude” of “mtDNA segregation”, it is hard to sort this out.

We apologise for the lack of clarity on this point. We have added more quantitative detail to this section of our results and addressed this point in the text.

6: The shift in LE oocytes and in LE pups is small but from the Fig 3B graph appears to be statistically significant (unless there has been a mistake in the error bar definition. See point 8 below.) No stats are reported for this, and the text description is just that they values are "comparable", which is open to interpretation. More detail is warranted here.

7: Figure 3. p values are given with no explanation of what is being tested or which statistical test is done. More detail must be given. These tests are complicated by the non-normal distributions of the heteroplasmy values, as they are strictly bounded by 0 and 1,

so it is important that the reader know what was done here to calculate those p-values. While I believe the spirit of these p-values is correct just by viewing the data plots in Fig 3A, the details do matter and should be given.

8: Fig 3B The Y axis says "mean and s.e.m.". Standard practice would be to use 2 x SEM for error bars, to approximate 95% confidence intervals. It is not clear to me whether this is an oversight in the axes label, or a mistake in the presentation of the error bar sizes.

9: Fig 3: The statistically significant difference between the heteroplasmy shift in the oocytes compared to the shift in the pups is rather important. Both the LE and HB models show significant differences, but in opposite directions. The meaning of this shift, and the difference in the shift between the two mouse models needs to be dealt with more fully than it is now. This is the most important result in the paper, and I am just left with the message that sometimes one thing happens and sometimes the opposite happens. Hardly enlightening.

Thanks to the reviewer for pointing out this lack of clarity in our analysis (6-8) and highlighting this interesting point (9). We have rephrased this results subsection and improved the corresponding figure to make the corresponding statistical tests and assumptions clear. We have highlighted the oocyte-pup shift and discussed this observation in more detail.

10: Line 287-290: This data on the distribution of p-values should be reported, at least in a supplementary figure.

We have included this plot and qq plots in the SI, and made clearer that we anticipate normality in the transformed, not the raw, heteroplasmy data.

11: Line 309-314: The authors discuss the extrapolated heteroplasmy variance at time 0, but do not actually report the value and confidence interval of that extrapolated number. The authors state that "the observed shift is of the same magnitude as identified by previous studies..." but the reader cannot judge that since the magnitude is never reported here.

Thanks for this spot – we have included this quantitative comparison.

12: Line 324-343: Model parameters are reported here with p values but with no confidence intervals. Please add confidence intervals throughout. The model parameters are shown in the plot and are reported throughout the text here. It would be convenient for the reader if these complete model parameters, along with confidence intervals, were also given in a simple supplementary table.

13: Line 339 "We do not find support for non-zero lambda". Based on Fig 6B, this looks borderline to me. Numerical values with confidence intervals and p values should really be given (either in text or a table) to support this statement.

We have added confidence intervals from bootstrapping throughout, including for the new human analysis.

14: Line 362 "and so set beta =0". This is incompatible with your data from both the HB and LE models which show significantly nonzero beta values. Why do a model calculation that is clearly inconsistent with the data presented in this paper?

Our original choice here was for simplicity: as the low (but significant) values of beta do not have a particularly strong effect on these distributions, we thought to keep the model

structure as simple as possible. But we appreciate that the more satisfactory approach is to include this parameter value: we have redone the calculations and corresponding plots to include the inferred beta values.

15: Lines 345-374 "Probability of observing heteroplasmy above or below a given value with age." This section is highly problematic. The interpretation given here is described as "in the context of preimplantation genetic diagnosis". The data presented in this paper shows strongly significant differences between the two mouse models, meaning that the parameter values derived from this data are certainly not universal for mammals, or even for mice.

Our idea here was to present an illustrative example of this predictive framework, which we fully appreciate must be parameterised to reflect the specific system under investigation. We have made this clearer in the text, and, as above, tightened up our claims about the directness of the comparison to human (or other) systems. We have adapted the analysis to be predictive rather than descriptive, using a training/test data separation to explore our model's predictive ability. We have also expanded this section to engage more directly with the human data from Rebolledo-Jaramillo et al., illustrating the extensibility of this framework to describe inheritance and drift patterns.

16: Lines 376-390 "Evidence for increasing heteroplasmy variance in somatic tissues." This short section is tacked on and does not really fit with the rest of the paper. The data here is from tissue samples, not single cell samples, and heteroplasmy variances measured at the single cell level and at the multi-cell level will be quite different just from basic math. Single cell variances will be much higher than multi-cell variances. The data is only partially given in Fig S2 and data from skin and spleen samples is mentioned in text but never presented to the reader. Either this section should be presented in full with complete data or it should be dropped. I suggest it be dropped as not truly relevant to the paper.

We appreciate this comment and have removed this section. We do hope to pursue this connection with somatic samples in future work, but appreciate that here it served to complicate the story.

17: Lines 478-480 "In this study we show that heteroplasmy variance increases continuously through time in organismal ageing. The traditional interpretation of this result would be that "bottleneck size" decreases with age..." No. The traditional interpretation would be that this is drift.

As above, we have engaged with this picture and removed this overly critical discussion of the "static bottleneck" approximation.

18: The fundamental result of this work is that mother-oocyte and mother-pup heteroplasmy shifts disagree and are even in statistically significantly opposite directions. Furthermore, the direction of that difference is opposite in the two mouse models. This fundamental result is not dealt with in sufficient detail in this paper. Possible reasons for these differences need to be discussed in more detail. It is particularly confusing to me as a reader that in Figure S1, which is supposed to give the reader a visual explanation of the possible dynamics of this system, presents 4 possibilities yet none of these correspond to the actual data in this paper showing opposite direction in mean in oocytes and pups. All four possibilities presented show the oocytes and pups behaving in exactly the same manner.

Thanks for this valuable suggestion. We have highlighted this result, discussed it further within the manuscript, and altered Fig S1 to include an illustration of the generational shift.

Minor points

19: This may be a minor point, but I found it quite distracting that the origin of the important label "LE" and "HB" was never given in the paper. As a reader, it left me wondering whether I had missed some important information in the meaning of these labels, or whether they labels were arbitrary. A simple statement of that at the first use of the LE and HB terms in the Methods would greatly improve the readability of this manuscript.

20: Please add "days" throughout when reporting mouse ages. Often the authors use "...mice at age 21." or similar phrases without units.

21: Line 450: (HB oocytes, Fig 2)". I believe they mean Fig 3.

Thanks for these points – we have addressed each of them.

Reviewer #2, an expert in statistical modelling (Remarks to the Author):

This is an interesting paper. However, the paper could be improved by considering the following specific points.

1. The abstract and highlights could be improved by including numerical examples.
2. The authors should explain exactly how the new findings might lead to improvements in understanding and prevention of genetic diseases. How might the findings be 'translated'? How could they have general applicability? Much more explanation is needed.
3. The statistical developments seem appropriate, but could be improved by much more use of numerical examples throughout.

Thanks to the reviewer for their positive perspective. We have included numerical examples in the abstract, and further supplementary plots to aid the reader's numerical intuition about these processes. In our added connection to human data we have expanded our discussion of the translational applicability of these results in the human system.

Reviewer #3, an expert in modelling (Remarks to the Author):

In this manuscript, the authors presented a study on the dynamics of mtDNA heteroplasmy in mice (two heteroplasmic mice HB and LE). Tissue samples from mothers (ear clip/tail) were taken at 21 day of age, and used as reference for quantifying changes in heteroplasmy in oocytes at different maternal age and in pups. The authors proposed a simple model of mtDNA heteroplasmy dynamics based on the assumption that mtDNA heteroplasmy follows a Gaussian distribution. The authors found that mtDNA heteroplasmy displays increased variance with age in germline and pups, as well as in somatic tissues. They also observed a mean shift in heteroplasmy between generations. Meanwhile, the heteroplasmy in germline appeared to diverge from the somatic precursors early in development.

Comments:

1. I found the study to be very timely and the data presented were impressive. The

observations and conclusions were supported by the data, if the model can be trusted (see below). The idea that mtDNA bottleneck sizes vary with age appears to suggest that the mtDNA bottleneck occurs postnatally. The timing of the bottleneck has indeed been under much debate (for example see Wai et al., Nat. Genet., 40, 1484–1488, 2008). A large body of studies, including a very recent paper from Chinnery lab (Floros et al, Nat. Cell Biol., 20, 144–151, 2018) put the bottleneck event during germline development. I hope the authors would expand the analysis of their dataset in the context of the timing of the bottleneck.

Thanks for this comment. We have added an introductory section introducing this branch of the literature and expanded our discussion of the bottleneck and associated processes governing mtDNA statistics during and after development. One point we want to get across is that it's both a “bottleneck” and an ongoing drift process that influence mtDNA variance – our findings that drift continues through life are not incompatible with an early developmental bottleneck reported by other studies. We have attempted to make this point clearer in our discussions.

2. The theoretical development of the model used in the analysis of the dataset needs more detail. As a mathematically-inclined person, I hope to see more consistent mathematical notation used in the manuscript and clear definition of the variables. For example, Δh was defined in M1 without arguments, but later in Eq. 1 and 2, have arguments that were never defined. Later on (in the paragraph before Eqn 3), Δh was described as Δh with two arguments (h, h_0). In addition, Eq. (3) has a square root function, but I could not be sure if the square root was for all terms in the bracket. I recommend the authors to use an equation editor to write their equations (e.g. Word Equation). In addition, Figure 2B is inconsistent with Eq. 1 and 2. The y-axis of Fig. 2B shows $h-h_{ref}$. Am I right to assume that the Gaussian distribution describes the difference in heteroplasmy h measured and a reference heteroplasmy h_{ref} . If so, then it is very hard to accept that Δh should be Gaussian.

Thanks for highlighting these issues. We have restructured the maths throughout the text and figures to keep the functional syntax of our model consistent. Our normality assumption corresponds to the transformed change in heteroplasmy. Raw heteroplasmy values and differences between them, as the reviewer implies, cannot be expected to be normally distributed, as they are constrained to lie on the interval between 0 and 1. Our transformation, however, maps this interval to the full real line and generally does yield reasonably normal distributions of mtDNA statistics (see point below). We have underlined this point in the main text.

3. I hope to see more support that the Gaussian distribution assumption is justifiable for their dataset. The non-rejection of Kolmogorov-Smirnov test needs to be supported by other tests of normalcy, e.g. using normal plot.

We have included a supplementary figure showing the distribution of KS p-values and qq plots for our transformed statistics and comparable samples from a normal distribution.

4. Looking at Figure 6 A alone, I am not entirely convinced that the model as proposed in the study, could explain the data. What were the p-values of the joint inference? How much was the variability in the data that could be explained by the model? Am I right to interpret the shaded area as 95% confidence interval? If so, a large number of data points are outside this region. Furthermore, looking at Figure 6 B, I can see that there is a possible parameter identifiability issue when fitting LE dataset, where the probability of

having parameters of different signs are not insignificant. The parameter inference for the HB dataset appeared better. While I am a proponent of keeping any model simple, I am not convinced that the model used in this study is able to explain the data.

We have addressed this important point in the description of the results and with a supplementary figure. Importantly, the intervals in Fig 6 are confidence intervals on the summary statistics of the data, which do not directly relate to the spread of the individual measurements. Fig S4 now shows the confidence intervals that correspond to the actual spread of heteroplasmy observations, which illustrates the ability of the increasing-variance model to capture the spread of heteroplasmy values through time. The p-values associated with the full model inference are reported in the text.

Reviewer #4, an expert in mitochondrial heteroplasmy (Remarks to the Author):

The authors present a large interesting dataset, and interpret this in the context of novel mathematical models. The main conclusion - of a maternal age effect on the level of heteroplasmy in the pups – has not been demonstrated before. It is, however, a very weak effect.

I agree that we should refer to heteroplasmy variance not bottleneck. However, I think the authors are adding to the confusion by mentioning a 'changing bottleneck', especially in the abstract. I appreciate that the authors are mathematicians, and that the bottleneck can be considered a theoretical construct leading to varied heteroplasmy levels. However, most biologists think in material terms, not the abstract. To them, the bottleneck is a biological process, so the 'bottleneck size' does not change with age. If I understand it correctly, the authors are really saying that the computed bottleneck size differs over time because the variance in heteroplasmy increases with age. This may be true – but the changing parameter here is the mathematical concept of bottleneck size, not necessarily the actual bottleneck itself, which has a biological basis. I think it would help to clarify this throughout the manuscript.

Thanks for this important point, which also aligns with the central comment of R1. We have reweighted our manuscript and rephrased several of our arguments accordingly. In particular, we now draw attention to the distinction between the developmental bottleneck and the ongoing process of genetic drift, and underline that we quantitatively characterise the contributions of these processes in temporal detail. We have added an introduction paragraph clarifying this point and replaced our discussion point (which proved confusing) with a more nuanced discussion of these points.

How do the authors account for the extreme outliers? In Fig 5, particularly the upper 2 panels, one of two data points have a very wide confidence interval. Why is this, and are these points influencing the weak signal they measure over time? I would like to see the analysis repeated without these unusual data points, to see if the reported relationships are robust.

The size of the confidence intervals for these points is due to the difficulty of sampling higher-order moments of distributions, like variance. The confidence intervals here are derived from the methodology of Wonnapijit et al. which focusses on accurately accounting for sampling errors in this context. We perform several classes of model fit in Fig 5, which account for this heterogeneity in uncertainty in different ways, and find that our results are robust with respect to particular choices of how to do this accounting. In particular, our findings are preserved when those datapoints with large confidence

intervals are penalised to reflect the increased associated uncertainty. We have made these points clearer in our description of this section.

Related to this, can the authors comment on the number of individual heteroplasmy measurements that are required to compute heteroplasmy variants. Samuels has done some work on this. I presume that there is a relationship between the confidence intervals and the number of measurements in each mouse/litter. Can they show that reduced numbers of measurements (eg in an ageing mouse, with lower fecundity) do not account for their observations?

Yes, we do employ the methods of Wonnapijit et al. (Samuels group) to quantify the sampling error in variance observations – see point above.

To my mind there is an inconsistency in the logic that needs to be addressed. If there is selection – as they observed – how can the distributions approximate a normal distribution? Surely it is skewed (ie with the top sliced off). How does this affect their results?

Thanks for identifying our lack of clarity here, also flagged by R3. Raw heteroplasmy distributions, even in the absence of selection, cannot be assumed to be normal, as they are constrained to lie between 0 and 1. However, the transformation that we employ maps this interval to the unconstrained real line, and we anticipate distributions under this transformation to be better approximated by a normal distribution. We have added confirmatory supplementary plots and a more detailed description in the main text to support this point.

A major finding is that different tissues behave differently over time. Can the authors show that their choice of sample (ear / tail biopsies) are not the 'cause' of the problem? Related to this, are biopsies done on the same day yielding the same result?

Previous work on these and other mouse models has shown that tail and skin (comparable to ear) tissue display negligible relative and absolute segregation shifts, especially in young mice. Animal welfare considerations prevent us from exploring the results of multiple biopsies on the same day; however, our model is reasonably robust with respect to experimental variability in heteroplasmy observations. We have made these points clearer in the ms.

It is disappointing that in the discussion they say that appropriate human datasets exist in the literature (line 508), yet they have not gone on to apply their formulae to this data - this would seem like an obvious and straightforward conclusion to the paper.

We're very grateful to the reviewer for this suggestion. Following it, we have included a new results subsection where we harness recent data from Rebolledo-Jaramillo et al. to illustrate the use of our model in the human context. While there is less data available here for model parameterisation, we find that our simple model describes and predicts patterns of heteroplasmy shift between generations, and gives inferred dynamic parameters consistent with a comparable recent theoretical study. We discuss how this work may be extended to give more detailed predictions with quantified uncertainty in future.

I also think that they must be more cautious in their predictions for humans. Some mutations eg m.3243 behave in a very unusual fashion, with the loss of mutation in blood. This demonstrates proof of principle that different mutations behave differently in different

tissues.

We have rephrased our description of the connection to the human system to avoid inappropriate claims of direct analogy. In the study of individual mutations where complex segregation patterns are known, our method could readily be applied on a tissue-by-tissue basis, and we are pursuing this line of research in somatic tissues in mice.

Minor:

line 339 – what is ‘non zero y’?

It should be clearer that the heteroplasmy variance data is indeed single-cell data in the oocytes, but the pup heteroplasmy variance is only measured on a pup-to-pup level, and not in single cells. In some cases they have made this distinction clear (e.g. in the figure 3 legend), but at other points it is implied that the pup heteroplasmy data is single-cell (e.g. line 264).

It would be nice to see a breakdown of the n-numbers in the supplementary data - i.e. 795 LE oocyte samples - how many females, age at harvesting for each animal, how many oocytes from each etc. Same for the pups. Perhaps this will already be submitted with the final paper

Thanks for these points – we have addressed them all in the text.

Reviewers' comments:

Reviewer #1 (Remarks to the Author):

The wording of the manuscript is much approved, and the authors have sufficiently responded to my primary concerns. However, their response and addition of new data has raised a new major concern.

Major concerns:

Line 437. The reported parameter values derived from the human dataset have huge confidence intervals, and in most cases even the sign of the parameters is unclear (with "0" lying within the confidence interval). This may arise from the limited size of the human dataset, or from the questionable decision to combine data from many different human pathogenic mutations into a single dataset. Many publications agree that the dynamics of heteroplasmy clearly differ across different pathogenic mutations (see PubMed ID 26740552 and 23390135, for example). The human data is dealt with too superficially to be of use in this paper. That question requires a much more thorough and careful analysis than is presented here, and it should be removed from this paper to focus on the detailed mouse data analysis.

Line 782: Figure 7: On the same topic, part B of this figure raises some issues. Why is the heteroplasmy threshold set to 0.1, which is very low? The format of the figure is confusing and it is not at all clear to me that the data does follow the model prediction. I don't find that this additional analysis (Fig 4B in particular) adds to the scientific value of this paper.

Minor concerns:

line 341: Not "in all cases". The HB pups appear to have a zero slope of heteroplasmy with time. Two fits are given to each of the 4 datasets (a naive fit and a weighted fit). Other than the p-values printed in the figures, I do not see the results of these 8 fits anywhere in this paper, other than this statement of "roughly 2.5×10^{-4} per day" here in the text. For a quantitative paper like this, this lack of reporting details of the results is not sufficient. The parameters from all 8 first should be presented in a table (supplementary is fine) and the text here should be modified to note the differences between the two mouse strains.

The same concern of lack of reporting of the model fit results holds for Figure 4. Again, supplementary tables reporting the parameters of these model fits should be included in the paper for a complete record of the results.

Reviewer #3 (Remarks to the Author):

The authors have satisfactorily addressed the majority of concerns that I had in their revised manuscript.

I have a few minor comments.

1. Paragraph 3 of Introduction: Besides random turnover, fusion-fission of mitochondria has been shown to affect mtDNA drift (see: Tam et al., PLoS Comput. Biol., 11: e1004183, 2015).
2. In Figure 1: Please correct typographical error; "genertions" should read "generations"
3. In Eqn. 3: Please clarify if the square-root is taken over 2 or over all terms following the square-root symbol. I raised this issue in one of my comments for the original manuscript.
4. Lines 437-440: The model parameters estimated for human data had confidence intervals that cross the zero axis. In other words, these parameters did not have strong statistical support, and

the true values may very well be 0. Could the authors clarify how robust the agreement between the data and their model prediction with respect to changes in the parameter perturbations? Would setting some of these parameters to 0 also produce the same model-data agreement? If so, this is a bit troubling.

Reviewer #4 (Remarks to the Author):

The authors have addressed my concerns

We are once more very grateful for the reviewer's positive comments on the manuscript. The only major change that was suggested was to remove the human data case study, which we have done. Other points are addressed below.

Reviewers' comments:

Reviewer #1 (Remarks to the Author):

The wording of the manuscript is much approved, and the authors have sufficiently responded to my primary concerns. However, their response and addition of new data has raised a new major concern.

Major concerns:

Line 437. The reported parameter values derived from the human dataset have huge confidence intervals, and in most cases even the sign of the parameters is unclear (with "0" lying within the confidence interval). This may arise from the limited size of the human dataset, or from the questionable decision to combine data from many different human pathogenic mutations into a single dataset. Many publications agree that the dynamics of heteroplasmy clearly differ across different pathogenic mutations (see PubMed ID 26740552 and 23390135, for example). The human data is dealt with too superficially to be of use in this paper. That question requires a much more thorough and careful analysis than is presented here, and it should be removed from this paper to focus on the detailed mouse data analysis.

Line 782: Figure 7: On the same topic, part B of this figure raises some issues. Why is the heteroplasmy threshold set to 0.1, which is very low? The format of the figure is confusing and it is not at all clear to me that the data does follow the model prediction. I don't find that this additional analysis (Fig 4B in particular) adds to the scientific value of this paper.

Thanks to the reviewer for these points. We introduced the human data case study in response to a suggestion from another reviewer and acknowledge the concerns that R1 raises. We attempted to be open about the possible shortcomings of this analysis (especially heterogeneity across different mutations) in our original wording of the corresponding results section, and about the fact that more targetted data would be required to optimise this analysis. However, we agree with the reviewers and editors that the manuscript is strong enough without the inclusion of this case study. In response to R1's and the editors' points in this iteration, we have removed this analysis and instead outlined the requirements and methodology that a rigorous application of our approach to human data would entail.

Minor concerns:

line 341: Not "in all cases". The HB pups appear to have a zero slope of heteroplasmy with time. Two fits are given to each of the 4 datasets (a naive fit and a weighted fit). Other than the p-values printed in the figures, I do not see the results of these 8 fits anywhere in this paper, other than this statement of "roughly 2.5×10^{-4} per day" here in the text. For a quantitative paper like this, this lack of reporting details of the results is not sufficient. The parameters from all 8 first should be presented in a table (supplementary is fine) and the text here should be modified to note the differences between the two mouse strains.

The same concern of lack of reporting of the model fit results holds for Figure 4. Again, supplementary tables reporting the parameters of these model fits should be included in the paper for a complete record of the results.

We have included this quantitative information as Supplementary Information.

Reviewer #3 (Remarks to the Author):

The authors have satisfactorily addressed the majority of concerns that I had in their revised manuscript.

I have a few minor comments.

1. Paragraph 3 of Introduction: Besides random turnover, fusion-fission of mitochondria has been shown to affect mtDNA drift (see: Tam et al., PLoS Comput. Biol., 11: e1004183, 2015).

2. In Figure 1: Please correct typographical error; "genertions" should read "generations"
3. In Eqn. 3: Please clarify if the square-root is taken over 2 or over all terms following the square-root symbol. I raised this issue in one of my comments for the original manuscript.
4. Lines 437-440: The model parameters estimated for human data had confidence intervals that cross the zero axis. In other words, these parameters did not have strong statistical support, and the true values may very well be 0. Could the authors clarify how robust the agreement between the data and their model prediction with respect to changes in the parameter perturbations? Would setting some of these parameters to 0 also produce the same model-data agreement? If so, this is a bit troubling.

Thanks for these points. We have addressed 1-3 through changes to the text and inclusion of the new citation. Point 4 is addressed by our removal of the human data case study.

Reviewer #4 (Remarks to the Author):

The authors have addressed my concerns

Thanks! In response to editorial comments and those from other reviewers, we have removed the human data case study as discussed, and replaced it with a description of how such a study will be performed in future. We remain very grateful for the suggestion of the human data inclusion, which we indeed plan to include in future work.